# De novo biosynthesis of antiarrhythmic alkaloid ajmaline

Jun Guo [1,5], Di Gao[2,5], Jiazhang Lian [2,3,4] ✉ & Yang Qu [1] ✉

The antiarrhythmic drug ajmaline is a monoterpenoid indole alkaloid (MIA) isolated from the Ayurvedic plant *Rauvolfia serpentina* (Indian Snakeroot). Research into the biosynthesis of ajmaline and another renowned MIA chemotherapeutic drug vinblastine has yielded pivotal advancements in the fields of plant specialized metabolism and engineering over recent decades. While the majority of vinblastine biosynthesis has been recently elucidated, the quest for comprehending ajmaline biosynthesis remains incomplete, marked by the absence of two critical enzymes. Here, we show the discovery and characterization of these two elusive reductases, alongside the identification of two physiologically relevant esterases that complete the biosynthesis of ajmaline. We show that ajmaline biosynthesis proceeds with vomilenine 1,2(*R*)-reduction followed by its 19,20(*S*)-reduction. This process is further modulated by two root-expressing esterases that deacetylate 17-*O*-acetylnorajmaline. Expanding upon the successful completion of the ajmaline biosynthetic pathway, we engineer the de novo biosynthesis of ajmaline in Baker's yeast.

The monoterpenoid indole alkaloid (MIA) ajmaline (Fig. 1) is a class Ia antiarrhythmic drug and used as a diagnostic agent for a rare type of arrhythmia Brugada syndrome[1,2]. First isolated in 1931, ajmaline is produced by extraction from the root of *Rauvolfia serpentina* (Indian Snakeroot), an Ayurvedic medicinal plant used in Hindu culture for hundreds of years under the names Sarpagandha (snake repellent) and Chandrá (moon, lunacy)[3]. Today, *R. serpentina* is valued clinically for both ajmaline and another antihypertensive MIA drug reserpine[4].

Ajmaline and >3000 other MIAs are all derived from the amino acid tryptophan and a monoterpenoid secologanin, which both contribute to their privileged, indole containing pharmacophores that give diverse bioactivities. Another well-known MIA drug is the chemotherapeutics vinblastine from *Catharanthus roseus* (Madagascar periwinkle). The over 30-step vinblastine biosynthesis has been recently elucidated, leading to partial and almost complete heterologous pathway reconstitution in several organisms[5–15]. The structural complexity of ajmaline and MIA in general has attracted generations of scientists for their total synthesis and biosynthesis. Recent work has demonstrated total chemical synthesis of ajmaline and related

sarpagan type MIAs[16–18]. However, ajmaline is still commercially extracted from *R. serpentina* and the biosynthesis of this classic drug remains to be completed.

After four decades, most of the ajmaline biosynthetic enzymes have been cloned and characterized. Similar to other MIAs, the biosynthesis of ajmaline begins with the condensation of tryptamine and the monoterpenoid secologanin by strictosidine synthase (STR) to form strictosidine, the central precursor to almost all MIAs (Fig. 1)[19]. Strictosidine deglycosylation by strictosidine β-glucosidase (SGD)[20] generates a series of iminium aglycones in spontaneous equilibrium, which are reduced by many reductases to numerous stereoisomers that feed into further biosynthetic pathways[21,22]. Among the many reductases, geissoschizine synthase (GS) generates 19*E*-geissoschizine[7], which is further oxidatively cyclized to polyneuridine aldehyde by a cytochrome P450 monooxygenase (CYP) sarpagan bridge enzyme (SBE)[23,24]. The carbomethoxy group from this unstable aldehyde is then removed by the polyneuridine aldehyde esterase (PNAE)[25]. The next enzyme vinorine synthase (VS) catalyzes a reversible acetylation reaction, while simultaneously forming an

[1]Department of Chemistry, University of New Brunswick, Fredericton, NB, Canada. [2]Key Laboratory of Biomass Chemical Engineering of Ministry of Education, College of Chemical and Biological Engineering, Zhejiang University, Hangzhou, China. [3]ZJU-Hangzhou Global Scientific and Technological Innovation Center, Zhejiang University, Hangzhou, China. [4]Zhejiang Key Laboratory of Smart Biomaterials, Zhejiang University, Hangzhou, China. [5]These authors contributed equally: Jun Guo, Di Gao. ✉e-mail: jzlian@zju.edu.cn; yang.qu@unb.ca

**Fig. 1 | The biosynthesis of ajmaline.** The enzymes in magenta are discovered and characterized in this study. STR strictosidine synthase, SGD strictosidine β-gluco-sidase, GS geissoschizine synthase, SBE1 and SBE2 sarpagan bridge enzyme, PNAE polyneuridine aldehyde esterase, VS vinorine synthase, VH vinorine hydroxylase, VR vomilenine 1,2(R)-reductase, DHVR 1,2-dihydrovomilenine 19,20(S)-reductase, AAE, AAE1, and AAE2 17-O-acetylnorajmaline acetyl esterase, NNMT norajmaline N-methyltransferase.

intramolecular cycle to produce vinorine[26,27]. Vinorine hydroxylation by another CYP vinorine hydroxylase (VH) gives vomilenine[28,29], which is further reduced stereospecifically by two reductases vomilenine reductase (VR, 1,2-reduction) and dihydrovomilenine reductase (DHVR, 19,20-reduction) to 17-O-acetylnorajmaline[30,31]. The terminal two steps involve the hydrolysis of the acetyl group by 17-O-acetylnorajmaline esterase (AAE)[32] and the indole N-methylation by nor-ajmaline N-methyltransferase (NNMT, Fig. 1)[33]. Ajmaline may also be further methylated by the ajmaline Nβ-methyltransferase (ANMT)[33]. Among these biosynthetic enzymes, VR and DHVR remain to be identified and characterized.

Previously, chromatographic partial purification from crude R. serpentina proteins from cell culture has led to the identification of the sequential VR and DHVR activities in planta[30,31]. Mass spectrometry on the partial purifications allowed the identification of several putative peptides of VR and DHVR[30,31]. With these short peptide sequences, Geissler et al. identified and cloned a reductase RsRR4 named as vomilenine reductase 2 that reduces the 19,20-double bond of vomilenine[34]. Contrary to the previous hypothesized reduction order, an alternative reduction order, namely 19,20-reduction followed by 1,2-reduction, may also exist in R. serpentina for ajmaline biosynthesis.

In this work, we complete the classic ajmaline biosynthetic path-way by identifying and characterizing both VR and DHVR, and two root-expressing AAE enzymes AAE1 and AAE2 from R. serpentina. We show that VR specifically performs 1,2-imine reduction and does not accept 19,20-dihydrovomilenine substrate. The previously character-ized RR4 is indeed a truncated variant of DHVR. While RR4/DHVR exhibits discernible vomilenine 19,20-reduction activity, its binding affinity towards vomilenine significantly lags behind that demon-strated for 1,2-dihydrovomilenine. This discrepancy arises from a substantial difference in $K_M$ values, with DHVR's $K_M$ for vomilenine being 34-fold greater compared to that observed for 1,2-dihy-drovomilenine. The AAE sequence originally cloned from R. serpentina cell culture is not present in two public sets of R. serpentina leaf and root transcriptomes. Instead, we clone and characterize two root-expressing enzymes AAE1 and AAE2, sharing 86% and 70% amino acid identity to AAE cloned from cell culture, and show their

acetylnorajmaline esterase activity. With these discoveries, we recon-stitute the complete biosynthetic pathway in Saccharomyces cerevisiae (Baker's yeast) for de novo ajmaline biosynthesis.

## Results
### Purification and identification of vomilenine from R. serpentina leaves
Since vomilenine is not commercially available, we examined the total alkaloids from R. serpentina grown in our greenhouse (Supplementary Fig. 1a, b) for the existence of this MIA. In both leaf and root tissues, we detected a major MIA with [M + H]⁺ m/z 351 expected from vomilenine, by the UV absorption profile (indolenine chromophore) and liquid chromatography tandem mass spectrometry (LC-MS/MS, Fig. 2a, Supplementary Figs. 1b and 2). In comparison, norajmaline and ajma-line were only found in the root tissue. After purifying the m/z 351 MIA from R. serpentina leaves with thin layer chromatography, we con-firmed that it was vomilenine by 1D/2D nuclear magnetic resonance (NMR) and by comparing the NMR chemical shifts to literature values[29,35,36] (Supplementary Figs. 3–7, Supplementary Table 1). To further confirm our identification, we co-expressed a previously characterized SBE from Gelsemium sempervirens[23], as well as PNAE, VS, and VH[25,26,29] from R. serpentina in yeast, and fed the strain with purified 19E-geissoschizine[7]. As expected from previous research, yeast co-expressing GsSBE, RsPNAE, and RsVS successfully produced a MIA with expected m/z 335 for vinorine (Fig. 2a). When we additionally co-expressed RsVH, we observed vomilenine accumulation instead of vinorine accumulation (Fig. 2a). Vomilenine produced in yeast showed identical LC-MS/MS retention time and fragmentation pattern with vomilenine purified from R. serpentina (Fig. 2a and Supplementary Fig. 2), which again confirmed our identification of vomilenine.

### Identification of vomilenine reductase candidates and addi-tional acetylnorajmaline esterase homologs
Previously the Stöckigt group has partially purified VR and DHVR from R. serpentina cell culture, which led to the identifications of several peptide fragments that might be part of these two enzymes[30,31]. Using these sequences, Geissler et al. identified and cloned two reductases

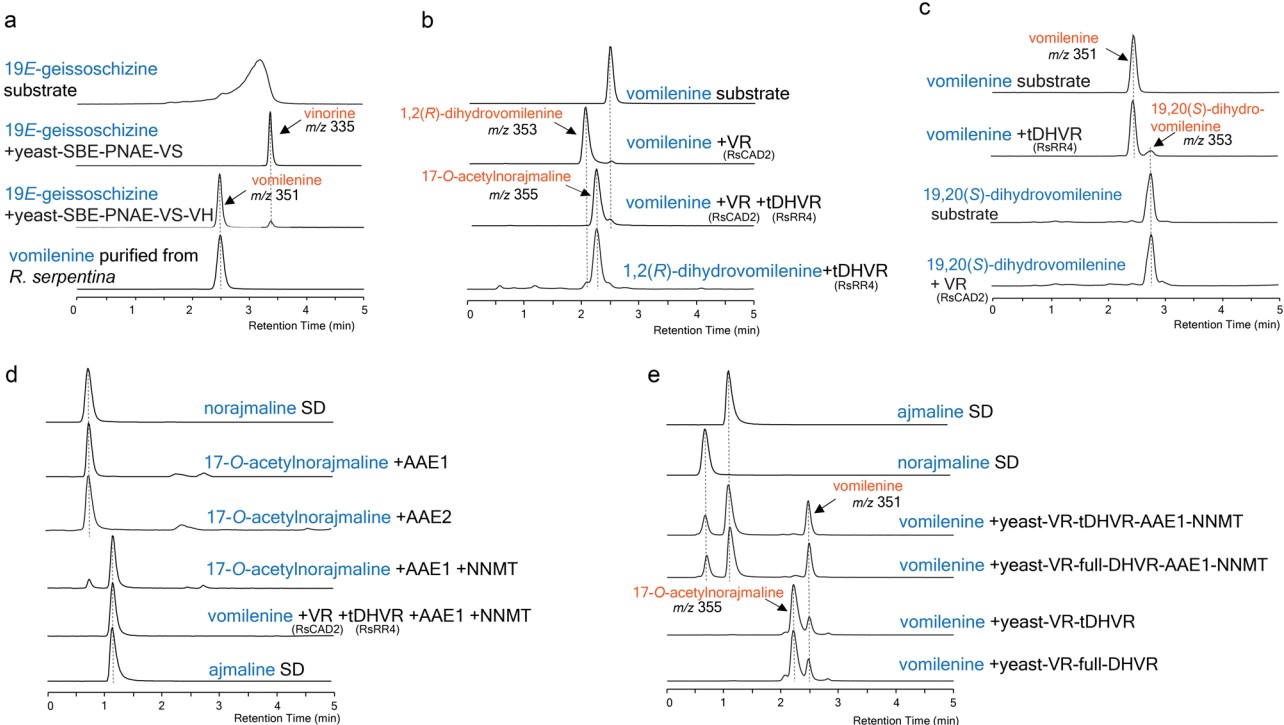

**Fig. 2 | Recombinant VR (RsCAD2), DHVR (RsRR4), and the root-expressing AAE1 and 2 enzymes complete the ajmaline biosynthesis. a** Yeast culture co-expressing *GsSBE*, *RsPNAE*, *RsVS*, and *RsVH* converted 19*E*-geissoschizine substrate to vomilenine that is identical to the vomilenine purified from *R. serpentina* leaves, while removing *RsVH* from the co-expression only led to vinorine formation. **b** Vomilenine was reduced to 1,2(*R*)-dihydrovomilenine by VR, which was further reduced by DHVR to 17-*O*-acetylnorajmaline. **c** DHVR showed detectable vomilenine 19,20(*S*)-reductase activity, while VR did not accept 19,20(*S*)-vomilenine substrate. **d** AAE1 and AAE2 enzymes that share 86% and 70% amino acid identity to the previously reported AAE from cell culture showed 17-*O*-acetylnorajmaline esterase activity to form norajmaline, allowing in vitro ajmaline biosynthesis by the activities of VR, DHVR, AAE1/2, and NNMT recombinant enzymes. **e** Yeast culture co-expressing VR, full or N-terminally truncated DHVR, AAE1, and NNMT converted vomilenine to ajmaline in vivo. The LC-MS/MS chromatograms show combined ESI-MRM ion transitions of [M + H]+ *m/z* 351→246, 353→82, 353→204, 355→80, 313→130, and 327→144 for detecting vomilenine, 1,2(*R*)-dihydrovomilenine, 19,20(*S*)-dihydrovomilenine, 17-*O*-acetylnorajmaline, norajmaline, and ajmaline, respectively for **b**, **c**, **d**, and **e**. For **a**, the chromatograms show the combined ESI-MRM ion transitions [M + H]+ *m/z* 335→258 and 351→246 for vinorine and vomilenine, respectively. Geissoschizine is shown by ESI-MRM ion transition [M + H]+ *m/z* 353→170. The ESI-MS/MS fragmentation patterns used to generate the MRM parameters are found in Supplementary Fig. 2.

from *R. serpentina* cell culture[34]. However, despite containing the VR peptide fragments, one enzyme RsRR6 turned out to be a bona fide cinnamyl alcohol dehydrogenase (CAD). The other enzyme RsRR4 showed 19,20-reductase activity with vomilenine substrate, which was thus named as vomilenine reductase 2[34]. Our group and other colleagues have previously characterized a number of CAD-like enzymes in MIA biosynthesis including geissoschizine synthase (GS), Redox1, demethylcorynantheidine synthase (DCS), heteroyohimbine synthase (HYS), Wieland–Gumlich aldehyde synthase (WS), and tabersonine 3-reductases (T3R) among others that are involved in reducing iminium/imine MIA intermediates[5–7,9,10,37–42]. We reasoned that VR and DHVR are likely members of the CAD-like reductases, and we searched a public *R. serpentina* leaf/root transcriptome (the PhytoMetaSyn project, https://bioinformatics.tugraz.at/phytometasyn/)[43] with the characterized CAD-like MIA reductases. We named the top 8 candidate reductases RsCAD1-8 based on their expression levels in *R. serpentina* root (Supplementary Table 2). We also identified RsGS, a homolog of RR6, and a homolog of RR4 among the top expressing CADs in root (Supplementary Table 2). The phylogenetic relationship among all characterized CAD-like MIA reductases is shown in Fig. 3.

We noticed gene sequence discrepancies between our identified sequences and previously reported sequences. We could not find the exact transcript of RR4[34] in PhytoMetaSyn transcriptome nor in another public *R. serpentina* transcriptome (Medicinal Plant Genomics Resources, MPGR, http://mpgr.uga.edu)[44]. In both datasets, the closest homolog was 95% identical to the reported RR4 among the aligned 363

amino acids, and this homolog also contained an additional 50 amino acids at the *N*-terminus (Supplementary Fig. 8). Similar discrepancies were noted for SBE, AAE, and RR6 (Supplementary Fig. 8). The SBE (thereafter named as RsSBE2, Genbank OQ591893) found in PhytoMetaSyn and MPGR transcriptomes was 95% identical to the reported amino acid sequence (thereafter named as RsSBE1, Genbank P0DO13)[23]. The originally reported AAE (Genbank Q3MKY2)[45] was not presented in these two transcriptomes either. Instead, we found three AAE homologs (AAE1-3), which had 86%, 70%, and 75% amino acid identity to the reported AAE (Supplementary Fig. 8, Supplementary Table 2). Additionally, the identified RR6 homolog (RR6-2) amino acid sequence was 95% identical to the reported sequence. In contrast, the sequences of the remaining pathway enzymes (GS, PNAE, VS, VH, and NNMT) in PhytoMetaSyn transcriptome were identical to those previously reported.

It is worth noting that the genes of RR4, RR6, and AAE were all cloned from *R. serpentina* cell cultures[34,45], while the two public transcriptomes were generated from mature plant leaf and root tissues. This difference may be the reason why different transcripts were found. To confirm the plant species, we cloned and sequenced the partial chloroplast 16 s rRNA gene *rps16* from our plant, which showed the expected 9 bp deletion and 8 bp insertion unique only to *R. serpentina* among 38 other Rauvolfia species (Supplementary Fig. 1c)[46]. The result indicated that correct species was used in our study. Since we did not have access to a *R. serpentina* cell culture, we cloned the PhytoMetaSyn/MPGR gene versions from *R. serpentina* plant. For RR4,

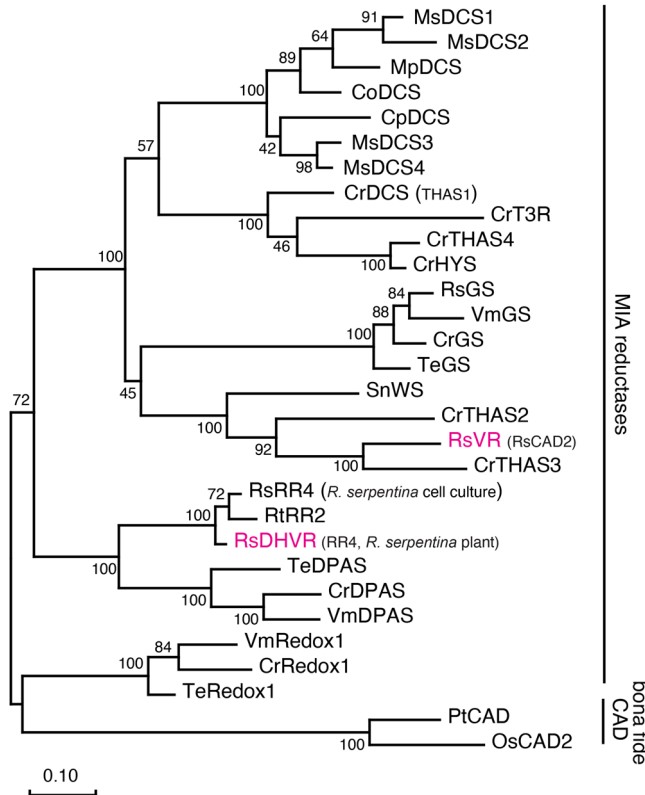

**Fig. 3 | Phylogenetic relationship of *R. serpentina* VR(CAD2) and DHVR(RR4) with other MIA imine/iminium reductases and alcohol dehydrogenases.** The enzymes labeled in magenta were investigated in this study. CAD cinnamyl alcohol dehydrogenase, VR vomilenine 1,2(*R*)-reductase, DHVR 1,2-dihydrovomilenine 19,20(*S*)-reductase, GS geissoschizine synthase, HYS heteroyohimbine synthase, THAS tetrahydroalstonine synthase, DCS dihydrodemethylcorynantheine/demethylcorynantheidine synthase, DPAS dihydroprecondylocarpine acetate synthase, Redox1 oxidized geissoschizine oxidoreductase 1, WS Wieland–Gumlich aldehyde synthase, T3R tabersonine 3-reductase. The plant species include: Cr *Catharanthus roseus*, Rs *Rauwolfia serpentina*, Rt *Rauwolfia tetraphylla*, Ms *Mitragyna speciosa*, Mp *Mitragyna parvifolia*, Co *Cephalanthus occidentalis*, Cp *Cinchona pubescens*, Vm *Vinca minor*, Te *Tabernaemontana elegans*, Sn *Strychnos nux-vomica*, Pt *Populus tremuloides*, Os *Oryza sativa*. The amino acid sequences and Genbank accession numbers are included in Supplementary Data 2. Evolutionary analyses were conducted in MEGA11. The evolutionary history was inferred by using the Maximum Likelihood method and JTT matrix-based model. The percentage of trees in which the associated taxa clustered together is shown next to the branches (100 bootstrap replicates). Initial tree(s) for the heuristic search were obtained automatically by applying Neighbor-Join and BioNJ algorithms to a matrix of pairwise distances estimated using the JTT model, and then selecting the topology with superior log likelihood value. The tree is drawn to scale, with branch lengths measured in the number of substitutions per site (scale bar).

we cloned both the full 413 amino acids and *N*-terminally truncated 363 amino acids (tRR4) versions. Based on the TargetP program (https://services.healthtech.dtu.dk/services/TargetP-2.0/), the extra amino acids did not contain a signal peptide. AAE was however predicted to contain a *N*-terminal secretory signal peptide, suggesting the requirement of eukaryotic secretory/glycosylation pathway for proper protein folding and perhaps post-translational modification. It was functionally expressed in *Nicotiana benthamiana* (tobacco) by agrobacterium mediated transient expression, while expression in *Escherichia coli* did not afford active enzymes[45]. We also identified *N*-terminal secretory signal peptides in RsAAE1-3 with TargetP program. Therefore, we opted to clone *RsAAE1-3* for expression in the eukaryotic yeast system, as prokaryotic *E. coli* might not recognize the signal peptide and accordingly generate functional enzymes.

## In vivo screening identified RsCAD2 as the vomilenine 1,2(*R*)-reductase

To test for VR activity, we expressed RsCAD1-8, RR4, tRR4, and RR6-2 in *E. coli*. We then fed the cells with purified vomilenine. The cells expressing RsCAD2 clearly reduced vomilenine (*m/z* 351) to an unknown dihydrovomilenine (*m/z* 353), which did not co-elute with 19,20-dihydrovomilenine (*m/z* 353) produced by either RR4 or tRR4 (Supplementary Fig. 9). The reduction was identified as 1,2-imine reduction, because the unknown dihydrovomilenine displayed typical UV absorption spectrum expected from MIAs with an indoline chromophore, such as ajmaline and norajmaline, in contrast to MIAs bearing an indolenine chromophore, such as vinorine, vomilenine, and 19,20-dihydrovomilenine (Fig. 1, Supplementary Fig. 2). However, the C2 stereochemistry could not be identified by UV spectroscopy. None of the remaining CAD candidates or RR6-2 showed vomilenine reduction activity. The RR4-mediated vomilenine 19,20-reduction appeared to be poor comparing to RsCAD2 activity (Supplementary Fig. 9).

## RsCAD2 and RsRR4 reduce vomilenine to 17-*O*-acetylnorajmaline

To further characterize RsCAD2 and RR4 activities, we purified the his-tagged enzymes from *E. coli* by Ni-NTA affinity chromatography (Supplementary Fig. 10a) and performed in vitro assays. As RR4 and tRR4 showed comparable in vivo activities (Supplementary Fig. 9) and tRR4 expressed markedly better in *E. coli* (Supplementary Fig. 10a), all the in vitro reactions were performed with tRR4. Consistent with *E. coli* feeding experiments, RsCAD2 reduced vomilenine to 1,2-dihydrovomilenine (*m/z* 353, Fig. 2b). With much lower activity, tRR4 could also reduce vomilenine to 19,20-dihydrovomilenine (*m/z* 353, Fig. 2c). When we assayed RsCAD2 and tRR4 in the same reaction, vomilenine was clearly further reduced to a tetrahydrovomilenine (*m/z* 355) with the same *m/z* value as 17-*O*-acetylnorajmaline (Fig. 2b). We obtained the same result when we used 1,2-dihydrovomilenine (*m/z* 353, RsCAD2 product) as substrate and only tRR4 in the reaction (Fig. 2c). In contrast, RsCAD2 did not accept 19,20-dihydrovomilenine (*m/z* 353) as substrate (Fig. 2c). As RR4 reduces the 19,20-double bond and RsCAD2 reduces the 1,2-double bond, it is likely that the tetrahydrovomilenine was 17-*O*-acetylnorajmaline.

## Root-expressing AAE enzymes allowed in vitro and in vivo biosynthesis of ajmaline

We reasoned that the conversion of this tetrahydrovomilenine to ajmaline would resolve its C2-stereochemistry. Therefore, we expressed RsAAE1-3 and NNMT that catalyze the last two steps for ajmaline biosynthesis. The *N*-terminal His-tagged NNMT was purified from *E. coli*[33]. We expressed the *C*-terminal His-tagged RsAAE1-3 individually in yeast, and fed the cultures with the tetrahydrovomilenine substrate, Yeast expressing RsAAE1 and RsAAE2 both converted it to norajmaline (*m/z* 313) when compared to an authentic norajmaline standard, while RsAAE3 did not show activity (Supplementary Fig. 11). When blotted with anti-his antibodies in a Western blot experiment, RsAAE1 showed ~10 kDa larger apparent molecular weight (Supplementary Fig. 10b), possibly due to glycosylation or other post-translational modification. RsAAE2 expression in yeast was too low to be detected by Western blot. Although recombinant RsAAE1 expressed in *E. coli* aligned with theoretical molecular weight (Supplementary Fig. 10c), the protein failed to catalyze the deacetylation reaction (Supplementary Fig. 11). Considering that RsAAE cloned from cell culture was not active when expressed in *E. coli*, the results collectively suggested that eukaryotic secretory pathway is required for functional expression of these esterases.

We then semi-purified RsAAE1 and 2 by Ni-NTA affinity chromatography (Supplementary Fig. 10b) and tested in vitro activity. Again, both recombinant RsAAE1 and 2 (86% and 70% amino acid identity to the AAE cloned from cell culture) hydrolyzed the

**Table 1 | Michaelis-Menten enzyme kinetics for RsVR and RsDHVR**

| Substrate | RsVR (RsCAD2) | | | RsDHVR (RsRR4, N-terminal truncated) | | |
|---|---|---|---|---|---|---|
| | $K_M$ (µM) | $k_{cat}$ (s⁻¹) | $k_{cat}/K_M$ (M⁻¹s⁻¹) | $K_M$ (µM) | $k_{cat}$ (s⁻¹) | $k_{cat}/K_M$ (M⁻¹s⁻¹) |
| Vomilenine | 41.6 | 1.70 | $4.09 \times 10^4$ | 1089 | 0.20 | 183 |
| 1,2-dihydrovomilenine | –[a] | – | – | 32.0 | 10.4 | $3.26 \times 10^5$ |
| 19,20-dihydrovomilenine | – | - | – | – | – | – |

Source data are provided as a Source Data file.

[a]Data not available as no enzyme activity was detected with the respective substrates.

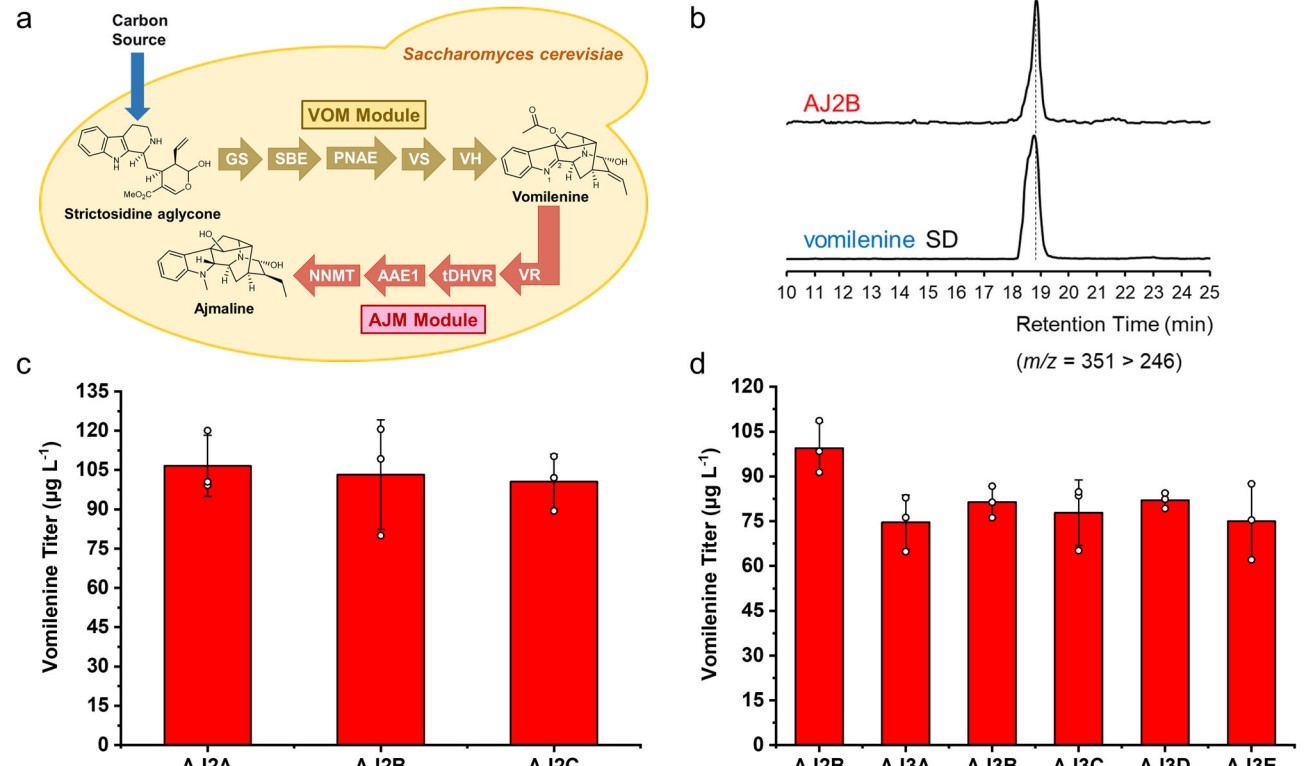

**Fig. 4 | Reconstitution of the VOM module for de novo biosynthesis of vomilenine. a** Schematic diagram for the metabolic pathway of ajmaline-producing strain. The downstream of the ajmaline biosynthetic pathway (from strictosidine aglycone to ajmaline) was divided into two functional modules. The golden was the VOM module (from strictosidine aglycone to vomilenine), including GS, SBE, PNAE, VS, and VH; the carmine was the AJM module (from vomilenine to ajmaline), including VR, DHVR or tDHVR, AAE1, and NNMT. **b** LC-MS analysis of de novo production of vomilenine in *S. cerevisiae*. MRM spectra (*m/z* 351 > 246) of vomilenine standard and AJ2B (vomilenine-producing strain) fermentation sample. **c** The vomilenine titer of vomilenine-producing strains AJ2A, AJ2B, and AJ2C with three

different versions of *SBE*. The strain AJ2A carried *RsSBE1* (Genbank P0DO13); AJ2B carried *GsSBE* (Genbank P0DO14); and AJ2C carried *RsSBE2* (Genbank OQ591893). Comparable titers of the three strains indicated similar catalytic activities of these three SBEs. **d** Increased copy number of pathway genes of the VOM module failed to increase vomilenine production. Strains AJ3A-AJ3E were constructed by integrating an additional copy of *GS, SBE, PNAE, VS,* and *VH* based on the strain AJ2B, respectively. The results represent the mean ± s.d. of biological triplicates (*n* = 3) and were graphed using OriginPro 2021 9.8.0.200 software. Source data are provided as a Source Data file.

tetrahydrovomilenine to norajmaline (Fig. 2d). When we included NNMT in the same reaction, ajmaline, also confirmed with an authentic standard, was produced at the expense of norajmaline (Fig. 2d). The results clearly showed that the tetrahydrovomilenine by the activities of RsCAD2/tRR4 was 17-*O*-acetylnorajmaline, and RsCAD2 was the expected vomilenine reductase (VR) that catalyzes the stereospecific vomilenine 1,2(*R*)-imine reduction. To further confirm our identification, we co-expressed RsCAD2, RR4 or tRR4, AAE1, and NNMT in yeast and tested their activities in vivo. Yeast co-expressing either RsCAD2/RR4 or RsCAD2/tRR4 combination produced 17-*O*-acetylnorajmaline when fed with vomilenine (Fig. 2e). Yeast co-expressing the four enzymes (either with RR4 or tRR4) produced ajmaline from vomilenine (Fig. 2e). The results again confirmed our identifications of these

enzymes and the stereochemistry of each intermediate. We therefore named RsCAD2 as vomilenine 1,2(*R*)-reductase (VR), and RsRR4 as the 1,2(*R*)-dihydrovomilenine 19,20(*S*)-reductase (DHVR). It is worth noting that RsVR does not contain the previously suggested peptide sequence from the partial purification experiment.

All four RsAAE primary sequences contained putative *N*-glycosylation sites as predicted by NetNGlyc-1.0 (https://services.healthtech.dtu.dk/services/NetNGlyc-1.0/) (Supplementary Fig. 8c). However, treating RsAAE1 with peptide-*N*-glycosidase F (PNGase F) that removes all types of *N*-glycans did not result in visible electrophoresis mobility difference, when compared to the untreated control. Both proteins however remained with higher apparent molecular weight in Western blot comparing to *E. coli* recombinant AAE1 (Supplementary Fig. 10c).

## Enzyme kinetics suggest that 1,2-vomilenine reduction precedes 19,20-reduction

The Stöckigt group suggested that vomilenine 1,2-reduction by VR occurs first, followed by the 19,20-reduction by DHVR. The hypothesis was based on the sequential enzyme activities from the partially purified enzymes from plant. To examine the reaction order for vomilenine 1,2-, and 19, 20-reductions, we studied the saturation kinetics of VR and the *N*-terminal truncated tDHVR (Table 1, Supplementary Fig. 12). While both enzymes accepted vomilenine as substrate, the $K_M$ of tDHVR (19, 20-reduction) was 26 times higher than that of VR (1,2-reduction). In contrast, tDHVR has much higher affinity to 1,2(*R*)-dihydrovomilenine, since its $K_M$ for vomilenine was 34 folds higher than that for 1,2(*R*)-dihydrovomilenine. Considering that VR did not accept 19,20-vomilenine substrate, our data strongly supported a sequential 1,2-reduction and 19,20-reduction, consistent with the studies by the Stöckigt group.

### Table 2 | List of genome-edited *S. cerevisiae* strains constructed in this study

| Strain | Genotype |
|---|---|
| AJ1 | AJM7-ΔHYS-IntG16::*GS*; IntG17::*RsPNAE-RsVS* |
| AJ2A | AJ1-IntG19::*RsVH*; IntG20::*RsSBE1* |
| AJ2B | AJ1-IntG19::*RsVH*; IntG20::*GsSBE* |
| AJ2C | AJ1-IntG19::*RsVH*; IntG20::*RsSBE2* |
| AJ3A | AJ2B-IntG24::*GS* |
| AJ3B | AJ2B-IntG24::*GsSBE* |
| AJ3C | AJ2B-IntG24::*RsPNAE* |
| AJ3D | AJ2B-IntG24::*RsVS* |
| AJ3E | AJ2B-IntG24::*RsVH* |
| AJ4A | AJ2B-IntL10::*RsAAE1-RsNNMT*; IntL12::*RsVR-DHVR* |
| AJ4B | AJ2B-IntL10::*RsAAE1-RsNNMT*; IntL12::*RsVR-tDHVR* |
| AJ5A | AJ4B-IntL7::*GS*; IntL9::*RsPNAE-RsVS* |
| AJ5B | AJ4B-IntL7::*GsSBE*; IntL9::*RsVR-tDHVR* |
| AJ5C | AJ4B-IntL7::*RsVH*; IntL9::*RsAAE1-RsNNMT* |
| AJ5D | AJ4B-IntL7::*RsPNAE-RsVS*; IntL9::*RsVR-tDHVR* |
| AJ5E | AJ4B-IntL7::*GS*; IntL9::*RsVR-tDHVR* |
| AJ5F | AJ4B-IntL7::*GsSBE*; IntL9::*RsAAE1-RsNNMT* |
| AJ5G | AJ4B-IntL7::*RsPNAE-RsVS*; IntL9::*RsVH* |
| AJ5H | AJ4B-IntL7::*RsVH*; IntL9::*RsVR-tDHVR* |
| AJ5I | AJ4B-IntL7::*GS*; IntL9::*RsAAE1-RsNNMT* |
| AJ5J | AJ4B-IntL7::*GsSBE*; IntL9::*RsPNAE-RsVS* |
| AJ6 | AJ5H-IntG21::*RsSBE1*; IntG22::*RsSBE2*; IntG4::*RsPNAE-RsVS*; IntG26::*RsPNAE-RsVS* |

The specific construction procedures of these strains were shown in Supplementary Fig. 13. The integration sites were detailed in Supplementary Table 4.

### De novo biosynthesis of vomilenine in yeast

After the identification and characterization of missing enzymes of the biosynthetic pathway, we aimed to achieve de novo biosynthesis of ajmaline. In our previous study, we have constructed a yeast strictosidine aglycone platform strain named AJM7-ΔHYS that could provide sufficient metabolic fluxes for downstream MIA biosynthesis[14]. This highly engineered strain included the iridoid biosynthetic genes, MIA biosynthetic genes, and other modifications that allowed the de novo biosynthesis of the MIAs catharanthine (527 µg L$^{-1}$) and vindoline (305 µg L$^{-1}$) after the remaining biosynthetic genes were integrated[14]. To facilitate pathway reconstitution and optimization, the ajmaline biosynthetic pathway (from strictosidine aglycone to ajmaline) was divided into two functional modules (Fig. 4a), the VOM module (from strictosidine aglycone to vomilenine) and the AJM module (from vomilenine to ajmaline).

In the strictosidine aglycone-producing AJM7-ΔHYS yeast strain[14], we first introduced the pathway genes of the VOM module (including *GS*, *SBE*, *PNAE*, *VS*, and *VH*). To explore the enzyme activities of various SBEs, we constructed three vomilenine-producing yeast strains, AJ2A with *RsSBE1* (Genbank P0DO13), AJ2B with *GsSBE* (Genbank P0DO14), and AJ2C with *RsSBE2* (Genbank OQ591893) (Table 2 and Supplementary Fig. 13). The fermentation results showed that all the three strains produced vomilenine de novo with comparable titers (~100 µg L$^{-1}$ in 48 h) (Fig. 4b, c). Although most pathway intermediates in the VOM

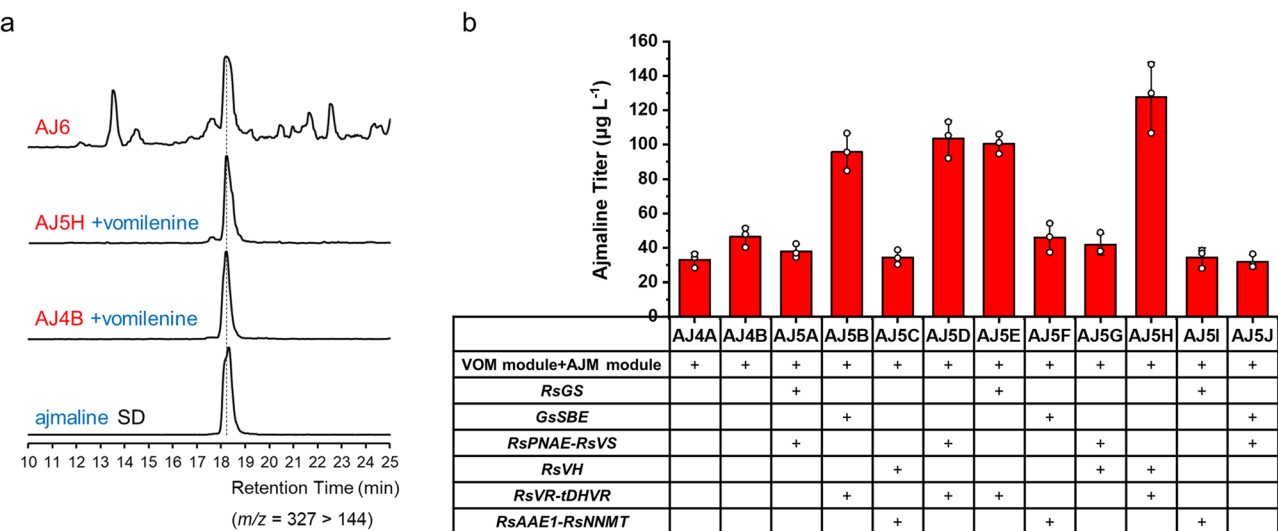

**Fig. 5 | Reconstitution and optimization of AJM module for de novo biosynthesis of ajmaline. a** LC-MS analysis of de novo production of ajmaline in strain AJ6 and the production of ajmaline by feeding 5 mg L$^{-1}$ vomilenine in strains AJ4B and AJ5H. MRM spectra (*m/z* = 327 > 144) of ajmaline standard and samples. **b** The ajmaline titer of different yeast strains with complete ajmaline biosynthetic pathway and extra copies of pathway genes, when feeding 5 mg L$^{-1}$ vomilenine. Additional copies of the genes from VOM module and AJM module were integrated into the genome combinatorially based on the strain AJ4B. Only the four strains (AJ5B, AJ5D, AJ5E, and AJ5H) with additional copies of *RsVR* and *tDHVR* showed significant increases in ajmaline production, indicating RsVR and tDHVR as the rate-limiting enzymes of the AJM module. The results represent the mean ± s.d. of biological triplicates (*n* = 3) and were graphed using OriginPro 2021 9.8.0.200 software. Source data are provided as a Source Data file.

module were unstable to be detected, we tried to identify and debottleneck the rate-limiting steps by integrating an additional copy of each VOM module gene into the genome of strain AJ2B. As shown in Table 2 and Supplementary Fig. 13, we constructed a series of strains AJ3A-AJ3E carrying an additional copy of *GS*, *SBE*, *PNAE*, *VS*, and *VH*, respectively. However, none of the strains showed significant improvement in vomilenine production (Fig. 4d). We suspected that vomilenine was not stable enough for continuous accumulation in yeast cells or the biosynthesis of vomilenine was rate-limited by multiple steps.

## De novo biosynthesis of ajmaline in yeast

For the de novo biosynthesis of ajmaline, we further introduced the AJM module genes (*VR*, *DHVR* or *tDHVR*, *AAE1*, and *NNMT*) into strain AJ2B, leading to the construction of AJ4A and AJ4B (Supplementary Fig. 13). Initial results showed that the two yeast strains (AJ4A with *DHVR* and AJ4B with *tDHVR*) carrying all the ajmaline biosynthesis pathway genes did not produce detectable ajmaline after fermentation. To verify the expressions of the AJM module genes, we collected yeast cells that had been induced by galactose for 24 hrs, resuspended the cells in Tris-HCl buffer (pH 8.0), and fed vomilenine at a final concentration of 5 mg L$^{-1}$. As shown in Fig. 5a, b, both AJ4A and AJ4B strains could successfully biosynthesize ajmaline with the feeding of vomilenine, and the titer in AJ4B (46 μg L$^{-1}$) was slightly higher than that in AJ4A (32 μg L$^{-1}$), indicating that the *N*-terminal truncation increased the enzymatic activity of DHVR in yeast. This result implied that all the AJM module genes were functionally expressed, and de novo biosynthesis of ajmaline was limited by the metabolic flux of the downstream pathway.

To increase the pathway metabolic flux and identify the bottleneck, we additionally introduced 10 combinations of downstream pathway genes into strain AJ4B. As shown in Table 2, Fig. 5b, and Supplementary Fig. 13, each gene combination consists of 3 or 4 genes from the VOM module and the AJM module, leading to the construction of yeast strains AJ5A-AJ5J. However, we still could not detect ajmaline production. When fed with vomilenine, four strains (AJ5B, AJ5D, AJ5E, and AJ5H) with additional copies of *RsVR* and *tDHVR* showed significant increases (AJ5B with 2.1-fold; AJ5D with 2.2-fold; AJ5E with 2.2-fold; and AJ5H with 2.8-fold) in ajmaline production (Fig. 5b). These results suggested that RsVR and tDHVR were the rate-limiting enzymes of the AJM module. The highest ajmaline titer was achieved by strain AJ5H at 128 μg L$^{-1}$ when fed with 5 mg L$^{-1}$ vomilenine. To further corelate vomilenine supply and ajmaline biosynthesis, we fed strain AJ5H with different concentrations of vomilenine, ranging from 5 to 5000 μg L$^{-1}$. As shown in Supplementary Fig. 14, ajmaline could be accumulated to significant levels when more than 250 μg L$^{-1}$ vomilenine was fed into the fermentation broth. On the other hand, with feeding of 100 μg L$^{-1}$ vomilenine, representing the equivalent amount of vomilenine generated by the VOM module in yeast, no significant accumulation of ajmaline was observed. These results indicated that the biosynthesis of ajmaline was limited by the availability of vomilenine.

Previous studies showed that the SBE product polyneuridine aldehyde and the PNAE product 16-epivellosimine were highly unstable[16,23,25,47–49]. Polyneuridine aldehyde in solution spontaneously oxidizes to a hemiacetal or degrades to an aromatized MIA flavopereirine[49], and in several cases its structural determination relied on quickly reducing it to the corresponding alcohol (polyneuridine)[16,17,47]. On the other hand, 16-epivellosimine spontaneously and rapidly epimerizes to its 16-epimer vellosimine, which could enter the sarpagine biosynthetic pathway[25,48]. We carefully analyzed the fermentation samples of AJ5H and detected the accumulation of tetrahydroalstonine (*m/z* 353, the by-product of GS, 83 μg L$^{-1}$) and other *m/z* 353 unknown by-products at high levels (Supplementary Fig. 15). We suspected that the instability of SBE and PNAE products

might have been responsible for decreased pathway flux and accumulation of by-products. When only GS and SBE were overexpressed in AJM7-ΔHYS yeast strain on multi-copy pESC vectors, putative polyneuridine aldehyde was detected in yeast culture, based on the reported MS/MS fragments[23]. However, the amounts lagged significantly behind other by-products such as THA and other unknown MIAs (Supplementary Fig. 16). When integrated a single copy of *GS*, *GS-GsSBE*, and *GS-GsSBE-PNAE* into the genome of strain AJM7-ΔHYS, LC-MS/MS profiles of these strains showed almost no difference, indicating no accumulation of polyneuridine aldehyde or the two known oxidation and degradation products (Supplementary Fig. 17). Thus, we additionally integrated two copies of *SBE*, *PNAE*, and *VS* into the genome of strain AJ5H to construct the final strain AJ6 (Table 2 and Supplementary Fig. 13), to strengthen the downstream pathway of 19*E*-geissoschizine and redirect metabolic flux from other by-products to our target product. LC-MS/MS spectrum in Fig. 5a showed the production of ajmaline with a titer of ~57 ng L$^{-1}$ in strain AJ6 from simple carbon sources. Western blot analysis using anti-cMyc antibodies confirmed the enhanced SBE and PNAE expressions in AJ6 strain (3 copies) when compared with its parental AJ2B strain (single copy) (Supplementary Fig. 18).

## Discussion

Over three decades have passed since the cloning of *R. serpentina STR* in 1988[50], which is not only the first gene cloned for the ajmaline biosynthetic pathway but also for all MIA biosynthesis. The Stöckigt group later made impressive characterizations of almost all ajmaline biosynthetic enzymes (SBE, PNAE, VS, VH, VR, DHVR, AAE) and established the biosynthetic order[24,25,27,28,30–32,45,48,51]. The group was also responsible for cloning *PNAE*, *VS*, and *AAE*, among other *R. serpentina* genes. The development of sequencing technology has made many plant transcriptomes and genomes available for MIA studies, and the discovery of over 20 biosynthetic genes in *C. roseus* in the past decade also helped speed up the gene discovery in *R. serpentina*. The recent cloning and characterization of GS, SBE, VH, and NNMT[7,9,23,29,33] allowed us to complete the entire pathway by identifying and characterizing the missing VR, DHVR, and two root-expressing AAE enzymes, for the biosynthesis of a classic, antiarrhythmic drug.

Our characterization suggests that vomilenine is first reduced by VR, then further reduced by DHVR to form 17-*O*-acetylnorajmaline. VR does not accept 19,20-dihydrovomilenine and only accepts vomilenine substrate (Fig. 2c). Although DHVR also shows vomilenine 19,20-reduction activity, DHVR clearly prefers 1,2-(*R*)-dihydrovomilenine with a 34-fold lower $K_M$ (affinity) for 1,2-(*R*)-dihydrovomilenine than vomilenine (Table 1). The identification of these intermediates and their stereochemistry has been supported by mass spectrometry, UV spectroscopy, and enzymatically converting them into ajmaline both in vitro and in vivo. VR in our study does not contain the peptides from partial purification experiment. RR6 cloned from cell culture previously and RR6-2 cloned in this study did not show VR activity despite containing these peptides (Supplementary Fig. 9). It is possible that RR6 or another protein containing these peptides co-purified with VR in previous purification from *R. serpentina* cell culture. Including VR and DHVR, there has been over 20 CAD-like reductases characterized for MIA biosynthesis in the past several years (Fig. 3). Despite bearing as low as 40% amino acid sequence identity, these diverse reductases evolved in four plant families (Apocynaceae, Rubiaceae, Gelsemiaceae, and Loganiaceae) in the order of Gentianales for MIA imine/iminium reductions[7,9,23,37,39,41,42], contributing to the impressive MIA diversification in nature. We anticipate more CAD-like reductases will be identified for the biosynthesis of other important MIAs.

Illumina transcriptome sequencing data showed that GS, VR and DHVR ranked the first, third, and fifth highly expressed CAD-like reductases in *R. serpentina* root (Supplementary Table 2). While GS was also the first highly expressed reductase in *R. serpentina* leaf for

vomilenine production, VR was not detected in leaf, and DHVR level was markedly lower in leaf than in root (Supplementary Table 2). Their expression profiles aligned with the fact that ajmaline accumulated only in root but not in leaf (Supplementary Fig. 1). While AAE1 expressed in both leaf and root, RsAAE2 was root specific (Supplementary Table 2), consistent with their role in ajmaline biosynthesis. The original AAE is likely only expressed in *R. serpentina* cell cultures and is not responsible for ajmaline biosynthesis in regular plants. It is possible that different variants are expressed in dedifferentiated cell suspension culture. A high-quality genome assembly is needed to locate the variants of AAE and DHVR and explain the sequence discrepancy that may be a result of gene duplication and selective expression profiles.

Both this study and previous study showed that functional expression of AAE enzymes requires eukaryotic cellular environment. All four AAE enzymes were predicted to contain the secretory signal peptides, which direct the nascent polypeptide trough the conserved eukaryotic secretory pathway[52]. The transiting polypeptides are usually *N*-glycosylated on the Asn-X-Ser/Thr sites, although the roles of such glycosylation are not clear. While *N*-glycosylation may be a reason for increased apparent molecular weights of AAE1 expressed in yeast in Western blot compared to recombinant protein expressed in *E. coli*, treating the enzyme with PNGase F that removes the *N*-glycans did not result in obvious mobility shift (Supplementary Fig. 10c). Peptide mass spectrometry may be able to detect the addition of glycans or other types of post-translational protein modifications that are responsible for the observed results.

After increasing the gene copy numbers of *SBE*, *PNAE*, *VS*, *VH*, *VR*, and *tDHVR*, we achieved de novo biosynthesis of ajmaline at a titer of $57 \, \text{ng} \, \text{L}^{-1}$ in *S. cerevisiae*. However, the titer was far from industrial applications, indicating the requirement of further pathway optimization using protein engineering and metabolic engineering approaches. Previously in our yeast AJM7-ΔHYS background that produces strictosidine aglycone, we recorded over $200 \, \mu\text{g} \, \text{L}^{-1}$ catharanthine production after single genomic integration of 8 catharanthine biosynthetic genes[14]. An extra genomic copy of *GS* more than doubled the catharanthine titer to over $500 \, \mu\text{g} \, \text{L}^{-1}$, suggesting that geissoschizine production was a major limiting factor for catharanthine titer[14]. In this same yeast background, we could only achieve ~$100 \, \mu\text{g} \, \text{L}^{-1}$ vomilenine with single genomic integration of 5 genes, and additional copies of *GS*, *SBE*, *PNAE*, *VS* or *VH* did not lead to any increase of vomilenine production (Fig. 4d). Western blot showed considerably higher SBE expression in AJ2C than AJ2B, however both strains accumulated comparable amounts of vomilenine (Fig. 4c and Supplementary Fig. 18). With the detection of only marginal amounts of polyneuridine aldehyde (Supplementary Figs. 16 and 17), our results suggested that the intermediate stability was one of major limiting factors for vomilenine production in yeast. Our finding is consistent with poor polyneuridine aldehyde and 16-epivellosimine stability in literatures[25,47,49] and the need of coupled SBE-PNAE-VS reaction to form the more stable intermediate vinorine[23]. However, we could not conclude whether these intermediates degrades spontaneously or they were converted by yeast native enzymes. In our de novo ajmaline producing strain AJ6, the highest accumulated MIA was tetrahydroalstonine ($63 \, \mu\text{g} \, \text{L}^{-1}$) and several unknown MIAs, which showed that most of ajmaline biosynthetic intermediates did not accumulate during continuous yeast fermentation (Supplementary Figs. 16 and 17). There may be auxiliary proteins or other mechanism in *R. serpentina* plant responsible for proper vomilenine accumulation. It is also possible to improve the biosynthesis by employing a scaffold-based pathway optimization strategy, which can co-localize the pathway enzymes to channel the pathway intermediates and minimize the loss of metabolic fluxes.

Downstream of vomilenine, both VR and tDHVR have moderate substrate affinity ($K_M$ $42 \, \mu\text{M}$ and $32 \, \mu\text{M}$ respectively, Table 1), which likely also contributed to the low metabolic flux, especially when vomilenine accumulated at ~$100 \, \mu\text{g} \, \text{L}^{-1}$ ($0.29 \, \mu\text{M}$) in yeast media. Feeding yeast with $5 \, \text{mg} \, \text{L}^{-1}$ ($14.5 \, \mu\text{M}$) vomilenine led to $46 \, \mu\text{g} \, \text{L}^{-1}$ ajmaline production indicated that high levels of vomilenine production was required for downstream ajmaline accumulation (AJ4B, Fig. 5b). We did not observe accumulations of any intermediates downstream of vomilenine, which suggested that these intermediates may not be stable in yeast environment and/or the VR and DHVR steps were rate limiting. Adding additional copies of both *VR* and *tDHVR* led to over 2-fold increase in ajmaline production to $96\text{-}128 \, \mu\text{g} \, \text{L}^{-1}$ (Fig. 5b, fed with $5 \, \text{mg} \, \text{L}^{-1}$ vomilenine), further confirmed that VR and DHVR were the rate limiting step in ajmaline biosynthesis in yeast. Considering all the results, ajmaline biosynthetic flux was hampered by multiple enzymatic steps in yeast. The de novo production of ajmaline in the final strain AJ6 is contributed by the increased expression levels of multiple enzymes, especially PNAE, VR, and tDHVR. It may be beneficial to engineer VR and DHVR or mining of their homologs in other genomes with better catalytic performance in subsequent studies.

In summary, our study documented the identification and characterization of four enzymes (VR, DHVR, and AAE1 and 2) that completed the decades-long investigation of the biosynthesis of ajmaline. The in vitro and in vivo biochemical characterization and root-specific expression of these enzymes supported their role in ajmaline production *in planta*. With the discoveries, we achieved the de novo ajmaline biosynthesis in a heterologous system and promised its large-scale production using synthetic biology approaches.

## Methods

### Chemicals
Ajmaline standards were purchased from Toronto Research Chemicals (Toronto, ON, Canada). Vinorine standard was purchased from Yuanye Bio-Technology Co. Ltd (Shanghai, China). Norajmaline standard was a generous gift from Dr. Vincenzo De Luca at Brock University[33]. 19*E*-Geissoschizine was produced by reacting strictosidine aglycones with purified geissoschizine synthase and purified by thine layer chromatography[7].

### Plant materials and RNA/cDNA synthesis
The plant *Rauwolfia serpentina* was grown in a glasshouse at 28 °C with 16/8 hrs photoperiod. A voucher specimen (UNB # 370328) has been deposited to the Connell Memorial Herbarium at the University of New Brunswick. Leaf (1 g) and root (1 g) tissues were collected for RNA extraction using standard TRIzol® RNA isolation reagent according to the manufacture's protocol (ThermoFisher Scientific). The resulting RNA was used to generate cDNA using the LunaScript®RT SuperMix Kit according to the manufacture's protocol (New England Biolabs).

### Vomilenine purification from plant material
Mature leaves (600 g) from glasshouse grown *R. serpentina* were harvested for vomilenine extraction. The leaves were submerged in ethyl acetate for 30 mins and evaporated. The alkaloids were extracted first by 1 M HCl and ethyl acetate. The aqueous phase was basified with NaOH to pH over 8, and subsequently extracted with ethyl acetate to afford total crude alkaloids. The crude alkaloids were then separated by TLC-silica gel 60 f254 (Sigma-Aldrich) with solvent ethyl acetate/methanol (9:1, v/v). TLC harvested vomilenine was identified by LC-MS/MS and NMR spectra.

### Cloning
Various ajmaline biosynthetic pathway genes were amplified from combined *R. serpentina* root and leaf cDNA. *RsCAD1-8* (Genbank OQ591881-591890) were amplified using primer sets (1–16), respectively. Full *DHVR* (Genbank OQ591882, PhytoMetaSyn version) and its N-terminal truncated *tDHVR* were amplified using primer sets (17/19, 18/19), respectively. *RsAAE1-3* (Genbank OQ591882, OR088065, OR088066) were amplified using C-terminus his-tagged primer sets

(20–25). *RsGS* (Genbank OQ591883), *RsSBE2* (Genbank OQ591893, PhytoMetaSyn version), *RsPNAE* (Genbank AF178576), *RsVS* (Genbank AJ556780), *RsVH* (Genbank KY926696), *RsNNMT* (Genbank KC708449) were amplified with primer sets (26–37), respectively. Codon-optimized *Gelsemium serpemvirens SBE* (*GsSBE* Genbank P0DO14) and *RsSBE1* (P0DO13, 95% identical to *RsSBE2*) were synthesized by Bio Basic Inc. (Markham, ON, Canada). The primers used in this study are listed in Supplementary Data 1.

For his-tagged protein purifications, RsCAD1,2, 4–8, and both full/truncated *RsDHVR* were cloned in pET30b(+) vector within *Bam*HI/*Sal*I sites. *RsCAD3* was cloned in pET30b(+) vector within *Eco*RI/*Sal*I sites. The full RsDHVR construct was mobilized to *E. coli* BL21A1 strain. The remaining *CAD* clones were mobilized to *E. coli* strain BL21DE3. For the glycoproteins, *RsAAE1* was cloned in pESC-URA yeast expression vector within *Eco*RI/*Not*I sites. *RsAAE2-3* were cloned in pESC-URA vectors within *Eco*RI/*Pac*I sites. Original Ruppert et al. AAE[45] was subcloned in pESC-URA vector by releasing the gene with *Eco*RI/*Not*I from pET28(+) vector and ligating it within the same sites. These yeast vectors were mobilized to *S. cerevisiae* BY4741 strain (MATα his3Δ1 leu2Δ0 met15Δ0 ura3Δ0 YPL154c::kanMX4).

For strictosidine aglycone to ajmaline pathway assembly in yeast, all the pathway genes were cloned into the multiple cloning sites (MCSs) of pESC series vectors by restriction/ligation (*Bam*HI/*Xho*I; *Bam*HI/*Sal*I; *Not*I) or Gibson Assembly. The recombinant plasmids harboring ajmaline biosynthetic pathway genes constructed in this study were listed in Supplementary Table 3.

## Recombinant protein expression and purifications
Overnight cultures (2 mL) of *E. coli* BL21DE3 strain harboring various plant genes in pET30b(+) vectors were used to inoculate 200 mL LB media, which were cultured at 200 rpm at 37 °C until $OD_{600}$ reached 0.6–0.7. The cultures were induced with 0.1 mM IPTG at 15 °C, 200 rpm overnight. For full DHVR, an overnight culture (2 mL) of *E. coli* strain BL21A1 was used to inoculate 200 mL LB media, which were cultured at 200 rpm and 37 °C until $OD_{600}$ reached 0.6–0.7. The cultures were induced with 0.1 mM IPTG and 0.1% (w/v) arabinose at 15 °C, 200 rpm overnight. The induced cells were sonicated in ice-cold sample buffer (20 mM Tris-HCl (pH 7.5), 100 mM NaCl, 10% (v/v) glycerol) supplemented with 20 mM imidazole, and purified using standard Ni-NTA affinity chromatography. After eluting with 250 mM imidazole in sample buffer, the purified recombinant proteins were desalted into sample buffer using a PD−10 desalting column (GE Health Sciences) according to manufacturer's protocol, and stored at −80 °C.

For AAE yeast expression and purification, an overnight culture (5 mL) of *S. cerevisiae* containing C-terminus His-tagged pESC-URA-*RsAAE1-3* were used to inoculate 500 mL of synthetic complete medium minus uracil (SC-Ura) with 2% glucose (w/v) overnight at 30 °C in a shaking incubator. The cells were collected by centrifugation (3000 × *g*, 2 min), washed twice with water, and resuspended in 500 mL SC-Ura medium with 2% galactose (w/v), which were induced for 24 h at 30 °C in a shaking incubator. The induced cells were collected, lysed in ice-cold sample buffer using a microtube homogenizer (Bead Bug, Bio Basic, Markham, ON, Canada) and glass beads (1 mm diameter) at 4 °C (30 Hz; 180 s per round for 3 rounds; 5 min pause in between rounds), and subsequently purified via standard Ni-NTA chromatography.

## SDS-PAGE and Western blot
Crude Yeast or *E. coli* proteins, or affinity-purified recombinant proteins were separated on 10% Acrylamide: Bis-Acrylamide 29:1 (40% Solution/Electrophoresis, ThermoFisher Scientific) SDS-PAGE and stained with Coomassie blue R-250. A non-stained gel was transferred to a PVDF membrane in a Bio-Rad Mini Trans-Blot Cell in transfer buffer (25 mM Tris-HCl, 192 mM glycine, 10% (v/v) methanol) at 50 v for 1 h.

After transfer, the gel was stained with Coomassie blue R-250, and the membrane was incubated in phosphate-buffered saline with Tween-20 (PBST, 137 mM NaCl, 2.7 mM KCl, 10 mM $Na_2HPO_4$, 1.8 mM $KH_2PO4$, 0.05% (v/v) Tween-20) for 1 h. The membrane was further incubated with His-Tag antibody (H-3, cat# sc-8036, mouse monoclonal, 200 μg mL$^{-1}$, Santa Cruz Biotechnology Inc., Dallas, Texas, USA) at dilution of 1:2000 in PBST or cMyc antibody (9E10, cat# sc-40, mouse monoclonal, 200 μg mL$^{-1}$, Santa Cruz Biotechnology Inc., Dallas, Texas, USA) at dilution of 1:3000 in PBST overnight. The membrane was washed three times, 20 min each time, in PBST, and further blotted with Donkey anti-Mouse IgG (H + L) Cross-Adsorbed Secondary Antibody, HRP (SA1−100, ThermoFisher Scientific) at dilution of 1:3000 in PBST overnight. The membrane was washed three times, 20 min each time, in PBST, and further developed with Amersham ECL Detection Reagents (Cytiva). The blot was visualized using a ChemiDoc Imaging Systems (Bio-Rad).

## In vitro assays and kinetics
A standard in vitro reaction (50 μL) included 20 mM Tris-HCl (pH 7.5), and some or all these components: 100 μM SAM, 1 mM NADPH, 1 μg of VR, tDHVR, and/or full DHVR, 1 μg RsAAE1, 1 μg RsNNMT, and 20 μg *R. serpentina* total root enzymes. The substrates included 0.2 μg vomilenine, 0.2 μg 1,2-dihydrovomilenine, 0.2 μg 19,20-dihydrovomilenine, and/or 0.2 μg 17-*O*-acetylnorajmaline. The reaction was incubated at 30 °C for 1 h and terminated by adding 150 μL methanol. The kinetics triplicated assays (50 μL) included 20 mM Tris-HCl (pH7.5), 1 mM NADPH, 1 μg VR or tDHVR, and substrate vomilenine at 4, 6.6, 10, 20, 30, 40, and 80 μM for VR, and 40, 66, 100, 200, 300, 400, and 800 μM for tDHVR. As well, tDHVR kinetics with substrate 1,2-dihydrovomilenine was measured at 4, 6.6, 10, 20, 30, 40, and 80 μM. The kinetics assays were performed at 30 °C for 2 min before they were terminated by adding 150 μL methanol to the reactions. The products were quantified using a standard curve to generate the enzyme velocity. The kinetics parameters and saturation curves were approximated using the software Prism 9.5.0 (GraphPad Software, LLC.).

## In vivo biotransformation
*E. coli* strains BL21DE3 or BL21A1 containing various ajmaline biosynthetic pathway genes were inoculated in 1 mL LB medium with appropriate antibiotics overnight at 37 °C in a shaking incubator. The overnight cultures were used to inoculate 10 mL fresh LB medium (1 in 100 dilution) with appropriate antibiotics, which were further grown at 37 °C in a shaking incubator until $OD_{600}$ reached 1.0. A final concentration of 0.1 mM IPTG (for BL21A1, 0.1% (w/v) arabinose) was then added to the cultures and induced overnight in a shaking incubator at 15 °C. The induced cells were collected and resuspended in 2 mL Tris-HCl (pH 7.5) supplemented with 10% (v/v) LB broth. Substrate including either 0.2 μg of vomilenine, 1,2-dihydrovomilenine, 19,20-dihydrovomilenine, or 17-*O*-acetylnorajmaline was added to the biotransformation mixture, which was incubated in a shaking incubator at 15 °C for 24 h. The cultures were mixed with equal volumes of methanol and filtered for LC-MS/MS analyses.

Yeasts containing various biosynthetic genes were inoculated in 1 mL drop-out (Leu, His, Ura, and/or Trp) SC medium with 2% glucose (w/v) overnight at 30 °C in a shaking incubator. The cells were collected by centrifugation, washed twice with water, and resuspended in 1 mL drop-out SC medium with 2% galactose (w/v) for 24 hrs at 30 °C in a shaking incubator. The induced cells were collected and resuspended in 1 mL Tris-HCl (pH 7.5) with 0.2 μg substrate vomilenine, 1,2-dihydrovomilenine, 19,20-dihydrovomilenine, 17-*O*-acetylnorajmaline, or 19*E*-geissoschizine. The biotransformation mixtures were incubated in a shaking incubator at 30 °C overnight, then equal volume of methanol was added for LC-MS/MS analyses.

## 1,2-dihydrovomilenine, 19,20-dihydrovomilenine, and 17-*O*-acetylnorajmaline enzymatic production

*E. coli* strains BL21DE3 containing *VR* or *tDHVR* in pET30b(+) vectors were inoculated in 2 mL LB medium overnight at 37 °C in a shaking incubator. The overnight cultures were used to inoculate 200 mL fresh LB medium with appropriate antibiotics, which were further grown at 37 °C in a shaking incubator until $OD_{600}$ reached 0.6. A final concentration of 0.1 mM IPTG was then added to the cultures and induced overnight in a shaking incubator at 15 °C. The induced cells were collected and resuspended in 20 mL Tris-HCl (pH 7.5) supplemented with 10% (v/v) LB. For 1,2-dihydrovomilenine production, 50 µg vomilenine was added to cells expressing VR. For 19,20-dihydrovomilenine production, 50 µg vomilenine was added to cells expressing tDHVR. For 17-*O*-acetylnorajmaline production, 50 µg vomilenine was added to an equal mixture of cells expressing VR and tDHVR. The reactions took place in a shaking incubator at 15 °C for 24 h, then it was extracted with 20 mL ethyl acetate. The evaporated extract was reconstituted in methanol and separated by thin layer chromatography using TLC-silica gel 60 f254 (Sigma-Aldrich) with solvent ethyl acetate:methanol (9:1, v-v).

## Construction of genome-edited *S. cerevisiae* strains

The genome modifications of *S. cerevisiae* strains were all based on the CRISPR/Cas9 system. Pathway gene cassettes with 40 bp homologous arms to the integration sites were obtained by PCR using the corresponding primers (primers 38–63 in Supplementary Data 1). The sgRNA plasmids were constructed by cloning the sgRNA spacer sequences designed by Benchling (https://benchling.com) into the *Bsa*I sites of pRS423-SpSgH and/or pRS426-SpSgH. Then 800 ng sgRNA plasmids and 800 ng gene cassettes with homologous arms (donor DNA fragments) were co-transformed into *S. cerevisiae* competent cells expressing SpCas9 by using the LiAc/SS carrier DNA/PEG method[53]. SED-URA/G418 or SED-HIS/G418 agar plates were used for the selection of recombinant yeast strains. The target strains were verified by colony PCR using the primers 64–89 and further confirmed by sequencing using primers 90–93 in Supplementary Data 1. The genome edited *S. cerevisiae* strains constructed in this study were listed in Table 2. The integration sites on *S. cerevisiae* genome were listed in Supplementary Table 4.

## Yeast fermentation

For yeast strains with complete vomilenine or ajmaline biosynthetic pathway, single colonies were selected from fresh YPD agar plates and inoculated into 3 mL YPD medium. After 24 hrs of cultivation at 250 rpm and 30 °C, 300 µL seed broth was inoculated into shake flask containing 30 mL YPD medium, and the cultivation was continued under the same conditions for another 24 h. Then yeast cells were collected by centrifugation at 4000 rpm for 5 min and washed twice with sterile water to remove residual glucose from the culture medium. All the yeast cells were then resuspended in 30 mL fresh YP medium with 2% galactose (w/v) and cultured at 250 rpm and 30 °C to induce the expression of exogenous genes. The yeast cells were collected by centrifugation, disrupted with glass beads (1 mm diameter), and extracted with equal volume of ethyl acetate. The upper layer (ethyl acetate solution) was used for qualitative and quantitative analysis after passing through a 0.22 µm membrane filter. For the detection of de novo produced ajmaline, the samples were concentrated 500 times using a freeze-dryer before LC-MS/MS measurements. All the fermentations were carried out with three biological replicates. The results were shown as the mean ± s.d. of biological triplicates. The bar and line graphs were generated with OriginPro 2021 9.8.0.200 software.

## NMR and LC-MS/MS

NMR spectra were recorded using an Agilent 400 MHz spectrometer with $CDCl_3$ referenced at 7.26 ppm. LC-MS/MS measurements were made using an Agilent Ultivo Triple Quadrupole LC-MS equipped with an Avantor Super C18 column (2.5 µm, 50 × 3 mm) and with the following solvent systems: solvent A, 29:71:2:398 (v/v) methanol:acetonitrile:1 M ammonium acetate:water; solvent B, 130:320:0.25:49.7 (v/v) methanol:acetonitrile:1 M ammonium acetate:water. The following linear gradient (8 mins, 0.6 mL min⁻¹) was used: 0 min, 20% solvent B; 0.5 min, 20% solvent B; 5.5 min, 99% solvent B; 5.8 min, 99% solvent B; 6.5 min, 20% solvent B; 8 min, 20% solvent B. The MS/MS was operated with gas temperature at 300 °C, gas flow of 10 L min⁻¹, capillary voltage 4 kV, fragmentor 135 V, collision energy 30 V with positive polarity. The Qualitative Analysis 10.0 software by Agilent was used for LC analyses.

When detecting samples from de novo biosynthesis strains, LC-MS/MS measurements were made using SHIMADZU Triple Quadrupole LC-MS/MS 8045 equipped with a HyPURITY C18 column (3.0 µm, 150 × 4.6 mm), for better separation of individual products with much more complex composition. The mobile phases were changed to the aqueous solution with 1‰ formic acid (solvent A) and methanol with 1‰ formic acid (solvent B). The gradient elution program (50 mins, 0.3 mL min⁻¹) was set as following: 0 min, 10% solvent B; 25 min, 90% solvent B; 35 min, 10% solvent B; and 50 min, 10% solvent B. The temperature of column oven was 30 °C; the atomizing gas flow rate was 3.0 L min⁻¹; the pressure of collision-induced dissociation (CID) gas was 230 kPa; the desolvation line (DL) temperature was 250 °C; and the temperature of heating block was 400 °C. The LabSolutions 5.91 software by SHIMADZU was used for data analyses of samples from de novo biosynthesis strains.

## Reporting summary

Further information on research design is available in the Nature Portfolio Reporting Summary linked to this article.

## Data availability

The primers are listed in Supplementary Data 1. The protein sequences for Fig. 3 are available in Supplementary Data 2. The RNA-seq for reads are available at NCBI-SRA ([https://www.ncbi.nlm.nih.gov/sra/?term=SRR26495086] and [https://www.ncbi.nlm.nih.gov/sra/?term=SRR26495085]). Source data are provided with this paper.

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

## Acknowledgements

This work was supported by National Key Research and Development Program (2021YFC2103200), the Natural Science Foundation of China (22278361), and Fundamental Research Funds for the Central Universities (226-2023-00015) grants to J.L., and the Natural Sciences and Engineering Research Council of Canada Discovery Grant (RGPIN-2020-04133) and New Brunswick Innovation Foundation Research Assistantship Initiative Grant (RAI_2021_067) to Y.Q.

## Author contributions

J.L. and Y.Q. conceived the research. J.G and D.G. performed the experiments. All authors wrote, revised, and approved the manuscript.

## Competing interests

The authors declare no competing interests.
