## [Peer Review File · Nature Communications]

Reviewers' Comments:

Reviewer #1:

Remarks to the Author:

What are the noteworthy results?

Guo et al identify the two missing enzymes for ajmaline biosynthesis, namely the vomilenine reductase (VR, 1,2-reduction) and dihydrovomilenine reductase (DHVR, 19,20-reduction) converting vomilenine to 17-O-acetyl norajmaline.

They also identify two unreported AAE enzymes AAE1 and AAE2 from *R. serpentina*.

Based on in vitro studies, synthetic biology, and analytical chemistry, the authors ultimately end up establishing a full de novo biosynthesis platform for ajmaline biosynthesis in yeast cell factories by increasing the gene copy numbers of SBE, PNAE, VS, VH, VR, and a truncated DHVR.

Will the work be of significance to the field and related fields? How does it compare to the established literature? If the work is not original, please provide relevant references.

The gene discovery and synthetic biology approach for refactoring MIA biosynthesis in yeast has been used for several other recent studies (including those of the Lian lab, Stöckigt lab, O'Connor lab, and Jensen/Keasling lab. As such, the methods are not novel. Still, the study does provide an interesting example of biosynthetic pathway gene discovery for a clinically relevant MIA.

Does the work support the conclusions and claims, or is additional evidence needed?

The conclusions are largely in line with the presented data, and the motivation for the experimental procedures sound. However, there are some speculations/hypothesis which would warrant verification:

- the hypothesized AAE1-glycosylation (L215). This should be possible to verify.
- The missing VR peptides from the partial purification experiment (L235).
- Intermediate stability being limiting for vomolenine biosynthesis is plausible. However, it would be of interest to the readership which intermediate would be limiting/destable. Could the authors try to truncate the pathway VOM module and perform LC-MS to verify this hypothesis (L369).

Are there any flaws in the data analysis, interpretation and conclusions? Do these prohibit publication or require revision?

No, there are no flaws in this study.

Is the methodology sound? Does the work meet the expected standards in your field?

Yes. The authors use a variety of approaches to characterise the enzymes. This reviewer would be interested to know (i.e. have the authors reason the choice) why *E. coli* was used for the reductase characterization, and then yeast for the esterase (AAE1-2) characterization.

Is there enough detail provided in the methods for the work to be reproduced?

Yes.

Minors:

L287/L300: Yield should not be presented as ug/L. This is the titer.

Reviewer #2:

Remarks to the Author:

The two plants *Rauwolfia serpentina* and *Catharanthus roseus* share not only the same family and the same group of specialized compounds – monoterpene indole alkaloids (MIA). They also have in common that they served over the last decades as subjects for numerous investigations on the biosynthesis of this important class of plant compounds. For *Rauwolfia*, the biosynthetic route leading to the antiarrhythmic ajmaline has been carefully analyzed on enzymatic level and – to a certain extent – also on the genetic level. But especially the responsible genes for late steps in biosynthesis remains elusive.

Other than *Catharanthus*, where MIAs are produced in the leaves, *Rauwolfia* is a slow-growing, woody plant where the main alkaloids appear predominantly in the roots. Since access to fresh plant material was limited, plant cell suspension cultures played an important role in the pathway elucidation. This circumstance together with the fact, that the Genus *Rauwolfia* encompasses numerous species, some of them hard to distinguish, challenges every effort to match the different datasets obtained from various genetic resources (why this is important I will line out later).

The presented manuscript now closes a gap in that it reports the first complete biosynthesis of ajmaline in a heterologous host. The authors meticulously identified candidate reductases and esterases to close the biosynthetic knowledge gap. On top of that, the engineered and optimized yeast strains to produce this very complex molecule. Although they share the fate with all groups being the first to produce a plant compound in an heterologous host: - the very low yield - it could be claimed that the pathway puzzle is solved with novel insights into the role of some old but also newly identified enzymes.

There are some issues I would like to point out which needs to be addressed:

The authors used plants grown in their lab for isolation of metabolites as well as RNA for subsequent cloning. What is needed is an unequivocal identifier of the plants, including a herbal specimen. Many sequences were compared to sequences obtained from PhytoMetaSyn (for which the website indicated is not responsive. Please provide primary literature) or from the medicinal plants consortium (which should be cited with the appropriate paper: Gongora-Castillo et al., PLOS One 2012). I would like to point out that also there, no proof is given that the *Rauwolfia* species was indeed *serpentina* (and not e.g. *R. tetraphylla*). This might be the reason that sequence deviations exist between various reported sequences derived from either *R. serpentina* plants, cell cultures or mistakenly as *serpentina* identified species. I recommend to provide sequencing of rps16 intron sequences as described in Eurlings et al. 2013, J. Forensic Sciences 58, 822-830.

Line 82: The statement “*R. serpentina* root proteins” is not in accordance with the papers cited under 30 + 31, as the two enzymatic activities were isolated from cell suspension cultures (same for line 124)

Line 173: the sole identification of a secretory signal does not justify the conclusion that glycosylation is “required”. It might just happen without consequence for the enzymatic activity. It could be easily checked if AAE-sequences contain N-glycosylation motifs (what they should as later on glycosylation is suggested by the higher apparent molecular size)

Line 288 ff: As I read, it ajmaline is only obtained from the AJM module if vomilenine is supplemented? Could the amount of vomilenine supplemented being related to the vomilenine produced by the previous modules? If there is a substantial gap then one can state that the metabolic flux is the issue.

Some minor points:

Line 29: capital letters for Indian Snakeroot (or not) should be harmonized throughout the manuscript (see also line 46) The Latin species name (*Rauwolfia serpentina*) should also be named prior to the common name.

Line 52: I agree that science is fascinating, but for the "bioactivities" another, more scientific adjective should be found

Line 53: It's "Madagaskar periwinkle" (without 's)

Line 183: I would suggest using the term "indoline chromophore" instead of "dihydroindole chromophore"

Line 212: the term "m/z 355 tetrahydrovomilenine" sounds strange, as there should be no tetrahydrovomilenine with another m/z. Also, there should be a period instead of a comma.

Line 259: "Module" should be lower case

Page 8, last paragraph: Authors should specify the yield of ajmaline in the strain AJ6 (ug/L) in the same manner as they did for other strains to facilitate comparison.

I suggest to include in Figure 1 the structure of unstable dialdehyde (in square parenthesis) to give the reader more information about this biosynthetic step since the authors mention: "... generates a series of iminium aglycones in spontaneous equilibrium, which are reduced by many reductases to numerous stereoisomers".

REVIEWER COMMENTS

Reviewer #1 (Remarks to the Author):

What are the noteworthy results?

Guo et al identify the two missing enzymes for ajmaline biosynthesis, namely the vomilenine reductase (VR, 1,2-reduction) and dihydrovomilenine reductase (DHVR, 19,20-reduction) converting vomilenine to 17-O-acetylajmaline.

They also identify two unreported AAE enzymes AAE1 and AAE2 from *R. serpentina*.

Based on in vitro studies, synthetic biology, and analytical chemistry, the authors ultimately end up establishing a full de novo biosynthesis platform for ajmaline biosynthesis in yeast cell factories by increasing the gene copy numbers of SBE, PNAE, VS, VH, VR, and a truncated DHVR.

Will the work be of significance to the field and related fields? How does it compare to the established literature? If the work is not original, please provide relevant references.

The gene discovery and synthetic biology approach for refactoring MIA biosynthesis in yeast has been used for several other recent studies (including those of the Lian lab, Stöckigt lab, O'Connor lab, and Jensen/Keasling lab). As such, the methods are not novel. Still, the study does provide an interesting example of biosynthetic pathway gene discovery for a clinically relevant MIA.

Does the work support the conclusions and claims, or is additional evidence needed?

The conclusions are largely in line with the presented data, and the motivation for the experimental procedures sound. However, there are some speculations/hypothesis which would warrant verification:

- the hypothesized AAE1-glycosylation (L215). This should be possible to verify.

We thank the reviewer for this point. The original AAE and the new AAE candidates AAE1-3 were all predicted to contain secretory signal peptides by the TargetP 2.0 program (DTU). We also ran the four AAE protein sequences through the NetNGlyc-1.0 (<https://services.healthtech.dtu.dk/services/NetNGlyc-1.0/>), which predicted putative Asn residues are presented in all four proteins for N-glycosylation. The results were updated in Supplementary Figure 8c.

Eukaryotic secretory pathway is quite conserved across single cellular and multicellular organisms, which involves translocation of nascent, unfolded polypeptide into ER lumen from the ribosome, proper protein folding with a number of ER lumen chaperons, and usually Asn N-glycosylation with mannose. These processes are involved in properly folding the protein and other post-translational modifications (e.g., disulfide bond formation). The folded proteins enter

Golgi after leaving ER lumen, where they may be further glycosylated with more sugars, which differs significantly in different lineages.

We have included the experiment expressing AAE1 in *E. coli*. The recombinant AAE1 in *E. coli* showed theoretical size in a Western blot, whereas the yeast recombinant AAE1 migrated significantly slower in the same blot (Supplementary Figure 10c). Also, the *E. coli* recombinant AAE1 was inactive when *E. coli* cells were fed with 17-*O*-acetylnorajmaline. Both the results suggested that AAE1 is only active in eukaryotic system but not in prokaryotes, which have different *N*-glycosylation system and likely not going to recognize eukaryotic secretory peptide. The large difference of apparent molecular masses between yeast and *E. coli* recombinant AAE1 is a strong indication of post-translational modification, although glycosylation could not be confirmed by just this experiment. Interestingly, treating yeast AAE1 with the glycosidase PNGase F that removes most N-glycans did not change the apparent AAE size in the blot. We updated the manuscript with these experiments and suggested that AAEs are likely modified by post-translational modification, but the exact modification is not clear and may be confirmed by peptide mass spectrometry (Line 250-255 of the revised manuscript).

- The missing VR peptides from the partial purification experiment (L235).

In both previous experiment by Geisler et al. (2016) and our experiment, the RR6 or RR6-2 protein containing precisely the “VR peptides” had no activity with vomilenine. Our best guess is that RR6/6-2 co-purified with VR in the partial purification experiment.

- Intermediate stability being limiting for vomolenine biosynthesis is plausible. However, it would be of interest to the readership which intermediate would be limiting/destable. Could the authors try to truncate the pathway VOM module and perform LC-MS to verify this hypothesis (L369).

We have expressed geissoschizine synthase (GS) and sarpagan bridge enzyme (SBE) in our *de novo* yeast strain AJM7- Δ HYS, on pESC-Ura plasmid. Very low amounts of putative polyneuridine aldehyde, the SBE product, was detected only when SBE was expressed. Since we do not have a polyneuridine aldehyde standard and this MIA was known to be unstable, its detection was based on 1) comparing the MS/MS fragmentation pattern to the reference by Dang et al. (2018) Nat Chem Biol, and 2) its appearance only when SBE was expressed. The result (Supplementary Figure 16) shows that only very small amounts of polyneuridine aldehyde accumulated in yeast culture, even when encoded on high-copy plasmid. Based on this experiment and the detection of ajmaline by increasing SBE, PNAE, and VS copies, we conclude that the intermediate stability is a limiting factor for vomilenine and ajmaline production in yeast. We briefly discussed the hypothesis in Line 336-340 of the revised manuscript.

Are there any flaws in the data analysis, interpretation and conclusions? Do these prohibit publication or require revision?

No, there are no flaws in this study.

Is the methodology sound? Does the work meet the expected standards in your field?

Yes. The authors use a variety of approaches to characterise the enzymes. This reviewer would be interested to know (i.e. have the authors reason the choice) why *E. coli* was used for the reductase characterization, and then yeast for the esterase (AAE1-2) characterization.

The reason for expressing AAE in yeast is the fact that these proteins are predicted to contain the secretory signal peptides. As far as we know, none of the secretory enzymes are successfully expressed in a prokaryotic system (e.g., Berberine Bridge Enzyme in benzyloquinoline alkaloid biosynthesis and tetrahydrocannabinolic acid synthase in cannabinoid biosynthesis). Previously our group discovered and characterized another putative glycoprotein O-acetylstemmadenine oxidase (ASO) in the MIA vinblastine biosynthetic pathway, which also contains secretory signal peptides. Its functional expression was only achieved in yeast and tobacco, while *E. coli* expression did not give functional proteins. We also confirmed that AAE1 expressed in *E. coli* was inactive (line 227-228) and Supplementary Figure 11.

With these prior experiences, we assumed that AAE could not be functionally expressed in *E. coli* and we opted to use yeast expression system. The rational and discussion on the use of eukaryotic system for AAE have been included in both the results and the discussion sections (Line 176-183 and 393-402 of the revised manuscript).

Is there enough detail provided in the methods for the work to be reproduced?

Yes.

Minors:

L287/L300: Yield should not be presented as $\mu\text{g/L}$. This is the titer.

We thank the reviewer for this point. We have changed “yield” to titer throughout the whole manuscript (such as Line 287, 307, 321, 408, and 412).

Reviewer #2 (Remarks to the Author):

The two plants *Rauwolfia serpentina* and *Catharanthus roseus* share not only the same family and the same group of specialized compounds – monoterpene indole alkaloids (MIA). They also have in common that they served over the last decades as subjects for numerous investigations on the biosynthesis of this important class of plant compounds. For *Rauwolfia*, the biosynthetic route leading to the antiarrhythmic ajmaline has been carefully analyzed on enzymatic level and – to a certain extent- also on the genetic level. But especially the responsible genes for late steps in biosynthesis remains elusive.

Other than *Catharanthus*, where MIAs are produced in the leaves, *Rauwolfia* is a slow-growing, woody plant where the main alkaloids appear predominantly in the roots. Since access to fresh plant material was limited, plant cell suspension cultures played an important role in the pathway elucidation. This circumstance together with the fact, that the Genus *Rauwolfia* encompasses numerous species, some of them hard to distinguish, challenges every effort to match the different datasets obtained from various genetic resources (why this is important I will line out later).

The presented manuscript now closes a gap in that it reports the first complete biosynthesis of ajmaline in a heterologous host. The authors meticulously identified candidate reductases and esterases to close the biosynthetic knowledge gap. On top of that, the engineered and optimized yeast strains to produce this very complex molecule. Although they share the fate with all groups being the first to produce a plant compound in an heterologous host: - the very low yield - it could be claimed that the pathway puzzle is solved with novel insights into the role of some old but also newly identified enzymes.

There are some issues I would like to point out which needs to be addressed:

The authors used plants grown in their lab for isolation of metabolites as well as RNA for subsequent cloning. What is needed is an unequivocal identifier of the plants, including a herbal specimen. Many sequences were compared to sequences obtained from PhytoMetaSyn (for which the website indicated is not responsive. Please provide primary literature) or from the medicinal plants consortium (which should be cited with the appropriate paper: Gongora-Castillo et al., PLOS One 2012). I would like to point out that also there, no proof is given that the *Rauwolfia* species was indeed *serpentina* (and not e.g. *R. tetraphylla*). This might be the reason that sequence deviations exist between various reported sequences derived from either *R. serpentina* plants, cell cultures or mistakenly as *serpentina* identified species. I recommend to provide sequencing of *rps16* intron sequences as described in Eurlings et al. 2013, J. Forensic Sciences 58, 822-830.

We thank the reviewer for the suggestion. An herbal specimen has been deposited to the Connell Memorial Herbarium at UNB with a voucher number UNB # 370328. We have also cloned and sequenced the chloroplast *rps16* intron and confirmed that it contained the signature indels that are indicative for *R. serpentina* (line 169-172 in the revised manuscript). The intron sequence and a photo of the plant have been included in the Supplementary Figure 1a and 1c.

The references to the PhytoMetaSyn project and the MPGR project have been both included in the text.

Unfortunately, the PhytoMetaSyn project website is currently not accessible, and we don't know when it will be brought back online.

Line 82: The statement “R. serpentina root proteins” is not in accordance with the papers cited under 30 + 31, as the two enzymatic activities were isolated from cell suspension cultures (same for line 124)

This has been corrected (Line 82 and 129 of the revised manuscript).

Line 173: the sole identification of a secretory signal does not justify the conclusion that glycosylation is “required”. It might just happen without consequence for the enzymatic activity. It could be easily checked if AAE-sequences contain N-glycosylation motifs (what they should as later on glycosylation is suggested by the higher apparent molecular size)

We thank the reviewer for pointing this out. Having a secretory signal peptide does not necessarily lead to protein glycosylation. We have revisited this issue by 1) identifying the putative Asn residues for *N*-glycosylation by NetNGlyc-1.0 (<https://services.healthtech.dtu.dk/services/NetNGlyc-1.0/>), 2) comparing the recombinant AAE sizes from *E. coli* and yeast in a Western blot, and 3) treating yeast recombinant AAE1 with PNGase F to remove the *N*-glycans from the Asn residues. The NetNGlyc program gave multiple putative Asn residues for glycosylation in all for AAE. The result is included in the updated Supplementary Figure 8c. Expressing AAE1 in *E. coli* gave the recombinant protein with correct size, but the enzyme was inactive (Supplementary Figure 11). This is consistent with Stöckigt's result when expressing the cell culture AAE1 in *E. coli*. It is intriguing that treating AAE1 with PNGase F did not obviously reduce its apparent molecular mass (Supplementary Figure 10c). We have included these experiments and suggested that 1) AAE functional expression could not be achieved in prokaryotic systems; 2) further experiments are needed to confirm the type of post-translational modification of AAE in yeast, which is responsible for the retardation in electrophoresis (line 176-183, 224-230, 250-255, and 393-402)

Line 288 ff: As I read, it ajmaline is only obtained from the AJM module if vomilenine is supplemented? Could the amount of vomilenine supplemented being related to the vomilenine produced by the previous modules? If there is a substantial gap, then one can state that the metabolic flux is the issue.

Thanks for the reviewer's suggestion. *De novo* biosynthesis of ajmaline was achieved in our finally engineered strain AJ6, without the supplementation of vomilenine. As for strain AJ4A and AJ4B mentioned by the reviewer here, ajmaline could be synthesized only when vomilenine was additionally supplemented.

To further address the reviewer's concern and better correlate vomilenine supply and ajmaline biosynthesis, we fed strain AJ5H with different concentrations of vomilenine, ranging from 5 to 5,000 $\mu\text{g L}^{-1}$. As shown in Supplementary Figure 15, ajmaline could be accumulated to significant levels when more than 250 $\mu\text{g L}^{-1}$ vomilenine was fed into the fermentation broth. On the other hand, with the fed of 100 $\mu\text{g L}^{-1}$ vomilenine, representing the equivalent amount of vomilenine generated by the VOM module in yeast, no significant accumulation of ajmaline was

observed. These results indicated that the biosynthesis of ajmaline was limited by the availability of vomilenine (Line 322-329 of the revised manuscript).

Some minor points:

Line 29: capital letters for Indian Snakeroot (or not) should be harmonized throughout the manuscript (see also line 46) The Latin species name (*Rauwolfia serpentina*) should also be named prior to the common name.

The changes have been made accordingly (Line 29 and 46 of the revised manuscript).

Line 52: I agree that science is fascinating, but for the “bioactivities” another, more scientific adjective should be found

We have deleted the adjective “fascinating” (Line 53 of the revised manuscript).

Line 53: It’s “Madagascar periwinkle” (without ‘s)

The change has been made (Line 54 of the revised manuscript).

Line 183: I would suggest using the term "indoline chromophore" instead of "dihydroindole chromophore"

The term has been changed in both main text and supplementary information (Line 191 of the revised manuscript).

Line 212: the term “m/z 355 tetrahydrovomilenine” sounds strange, as there should be no tetrahydrovomilenine with another m/z. Also, there should be a period instead of a comma.

We have removed m/z 355 as suggested (Line 212 of the revised manuscript).

Line 259: “Module” should be lower case

We have changed all the terms to lower case (such as Line 279, 280, 283, 289, and 291).

Page 8, last paragraph: Authors should specify the yield of ajmaline in the strain AJ6 ($\mu\text{g/L}$) in the same manner as they did for other strains to facilitate comparison.

We have calculated and provided the titer of ajmaline ($\sim 57 \text{ ng L}^{-1}$) in strain AJ6 (Line 344 of the revised manuscript).

I suggest to include in Figure 1 the structure of unstable dialdehyde (in square parenthesis) to give the reader more information about this biosynthetic step since the authors mention: "... generates a series of iminium aglycones in spontaneous equilibrium, which are reduced by many reductases to numerous stereoisomers".

Thanks for the reviewer’s suggestion. Figure 1 has been updated to include the aglycone structures.

Reviewers' Comments:

Reviewer #1:

Remarks to the Author:

Qu et al., response to author rebuttal:

Majors:

1. the hypothesized AAE1-glycosylation (L215). This should be possible to verify.

>> We thank the reviewer for this point. The original AAE and the new AAE candidates AAE1-3 were all predicted to contain secretory signal peptides by the TargetP 2.0 program (DTU). We also ran the four AAE protein sequences through the NetNGlyc-1.0

(<https://services.healthtech.dtu.dk/services/NetNGlyc-1.0/>), which predicted putative Asn residues are presented in all four proteins for N-glycosylation. The results were updated in Supplementary Figure 8c. Eukaryotic secretory pathway is quite conserved across single cellular and multicellular organisms, which involves translocation of nascent, unfolded polypeptide into ER lumen from the ribosome, proper protein folding with a number of ER lumen chaperons, and usually Asn N-glycosylation with mannose. These processes are involved in properly folding the protein and other post-translational modifications (e.g., disulfide bond formation). The folded proteins enter Golgi after leaving ER lumen, where they may be further glycosylated with more sugars, which differs significantly in different lineages. We have included the experiment expressing AAE1 in *E. coli*. The recombinant AAE1 in *E. coli* showed theoretical size in a Western blot, whereas the yeast recombinant AAE1 migrated significantly slower in the same blot (Supplementary Figure 10c). Also, the *E. coli* recombinant AAE1 was inactive when *E. coli* cells were fed with 17-O-acetyl norajmaline. Both the results suggested that AAE1 is only active in eukaryotic system but not in prokaryotes, which have different N-glycosylation system and likely not going to recognize eukaryotic secretory peptide.

The large difference of apparent molecular masses between yeast and *E. coli* recombinant AAE1 is a strong indication of post-translational modification, although glycosylation could not be confirmed by just this experiment. Interestingly, treating yeast AAE1 with the glycosidase PNGase F that removes most N-glycans did not change the apparent AAE size in the blot. We updated the manuscript with these experiments and suggested that AAEs are likely modified by post-translational modification, but the exact modification is not clear and may be confirmed by peptide mass spectrometry (Line 250-255 of the revised manuscript).

Reviewer: I appreciate the in silico efforts and attempts to functionalize AAE1 in *E. coli*. The further experimental testing of the glycosylation hypothesis as well as the updated manuscript text on the intriguing finding of the molecular mass of AAE when expressed in yeast improves the credibility and the response is satisfactory.

- The missing VR peptides from the partial purification experiment (L235).

>> In both previous experiment by Geisler et al. (2016) and our experiment, the RR6 or RR6-2 protein containing precisely the "VR peptides" had no activity with vomilenine. Our best guess is that RR6/6-2 co-purified with VR in the partial purification experiment.

Reviewer: This is plausible, and response considered sufficient.

- Intermediate stability being limiting for vomilenine biosynthesis is plausible. However, it would be of interest to the readership which intermediate would be limiting/destable. Could the authors try to truncate the pathway VOM module and perform LC-MS to verify this hypothesis (L369).

>> We have expressed geissoschizine synthase (GS) and sarpagan bridge enzyme (SBE) in our de novo yeast strain AJM7- Δ HYS, on pESC-Ura plasmid. Very low amounts of putative polyneuridine aldehyde, the SBE product, was detected only when SBE was expressed. Since we do not have a polyneuridine aldehyde standard and this MIA was known to be unstable, its detection was based on 1) comparing the MS/MS fragmentation pattern to the reference by Dang et al. (2018) *Nat Chem Biol*, and 2) its appearance only when SBE was expressed. The result (Supplementary Figure 16) shows that only very small amounts of polyneuridine aldehyde accumulated in yeast culture,

even when encoded on high-copy plasmid. Based on this experiment and the detection of ajmaline by increasing SBE, PNAE, and VS copies, we conclude that the intermediate stability is a limiting factor for vomilenine and ajmaline production in yeast. We briefly discussed the hypothesis in Line 336-340 of the revised manuscript.

Reviewer: The AJM7-deltaHYS strain from the authors' previous study (orig. ref. 14) is indeed a relevant departure point for this study. However, there are several plausible reasons for the lack of increased vomilenine abundance beyond intermediate instability. While, it is true that Dang et al., reported low levels of polyneuridine aldehyde and 16-epivellosimine in yeast, in both Dang et al., and in this study, such low abundance could equally well be caused by shunt product formations derived from native yeast enzymes. Likewise, because genes encoding plant biosynthetic enzymes are expressed from high-copy plasmids in yeast, does not mean that these enzymes indeed will express well in yeast. Such logic should be avoided, and/or substantiated by (relative) quantification of enzyme levels (e.g. tagged-protein expression, targeted proteomics). Also, the hypothesis that "the yeast environment" is suboptimal for the stability of intermediates between geissoschizine and ajmaline is an overstatement without further analysis on this hypothesis. Further to this, this reviewer does not see that over-expression analysis of SBE, PNAE, and VS leads to increased ajmaline biosynthesis - this is only obtained by over-expression of VR-DHVR (Fig. 5B),

In conclusion, this reviewer does not find the argumentation for the low vomilenine abundance to be fully justified based on Dang et al., and by over-expression analysis of SBE, PNAE, and VS, and would like to see more experimental evidence before this hypothesis can stand alone. Alternatively, but less justified considering the target journal/audience, would be to discuss further the observed limited abundances of vomilenine by considering the alternative reasoning from this reviewer.

Are there any flaws in the data analysis, interpretation and conclusions? Do these prohibit publication or require revision?

No, there are no flaws in this study.

Is the methodology sound? Does the work meet the expected standards in your field?

Yes. The authors use a variety of approaches to characterise the enzymes. This reviewer would be interested to know (i.e. have the authors reason the choice) why *E. coli* was used for the reductase characterization, and then yeast for the esterase (AAE1-2) characterization.

>> The reason for expressing AAE in yeast is the fact that these proteins are predicted to contain the secretory signal peptides. As far as we know, none of the secretory enzymes are successfully expressed in a prokaryotic system (e.g., Berberine Bridge Enzyme in benzylisoquinoline alkaloid biosynthesis and tetrahydrocannabinolic acid synthase in cannabinoid biosynthesis). Previously our group discovered and characterized another putative glycoprotein O-acetylstemmadenine oxidase (ASO) in the MIA vinblastine biosynthetic pathway, which also contains secretory signal peptides. Its functional expression was only achieved in yeast and tobacco, while *E. coli* expression did not give functional proteins. We also confirmed that AAE1 expressed in *E. coli* was inactive (line 227-228) and Supplementary Figure 11. With these prior experiences, we assumed that AAE could not be functionally expressed in *E. coli* and we opted to use yeast expression system. The rationale and discussion on the use of eukaryotic system for AAE have been included in both the results and the discussion sections (Line 176-183 and 393-402 of the revised manuscript).

Reviewer: The reviewer thanks the authors for the clarification and regards the manuscript update satisfactory. If the authors hold negative data for functional *E. coli* expression of AAEs it would be beneficial for the readers to have it inserted in the manuscript, e.g. Suppl. Fig. 11.

Is there enough detail provided in the methods for the work to be reproduced?

Yes.

Minors:

L287/L300: Yield should not be presented as $\mu\text{g/L}$. This is the titer.

>> We thank the reviewer for this point. We have changed "yield" to titer throughout the whole

manuscript (such as Line 287, 307, 321, 408, and 412)

Reivewer: Good.

Reviewer #2:

Remarks to the Author:

The authors have sufficiently addressed all suggestions and comments. I recommend publication.

REVIEWER COMMENTS

Reviewer #1 (Remarks to the Author):

Qu et al., response to author rebuttal:

Majors:

1. the hypothesized AAE1-glycosylation (L215). This should be possible to verify.

We thank the reviewer for this point. The original AAE and the new AAE candidates AAE1-3 were all predicted to contain secretory signal peptides by the TargetP 2.0 program (DTU). We also ran the four AAE protein sequences through the NetNGlyc-1.0 (<https://services.healthtech.dtu.dk/services/NetNGlyc-1.0/>), which predicted putative Asn residues are presented in all four proteins for N-glycosylation. The results were updated in Supplementary Figure 8c.

Eukaryotic secretory pathway is quite conserved across single cellular and multicellular organisms, which involves translocation of nascent, unfolded polypeptide into ER lumen from the ribosome, proper protein folding with a number of ER lumen chaperons, and usually Asn N-glycosylation with mannose. These processes are involved in properly folding the protein and other post-translational modifications (e.g., disulfide bond formation). The folded proteins enter Golgi after leaving ER lumen, where they may be further glycosylated with more sugars, which differs significantly in different lineages.

We have included the experiment expressing AAE1 in *E. coli*. The recombinant AAE1 in *E. coli* showed theoretical size in a Western blot, whereas the yeast recombinant AAE1 migrated significantly slower in the same blot (Supplementary Figure 10c). Also, the *E. coli* recombinant AAE1 was inactive when *E. coli* cells were fed with 17-*O*-acetylnorajmaline. Both the results suggested that AAE1 is only active in eukaryotic system but not in prokaryotes, which have different N-glycosylation system and likely not going to recognize eukaryotic secretory peptide. The large difference of apparent molecular masses between yeast and *E. coli* recombinant AAE1 is a strong indication of post-translational modification, although glycosylation could not be confirmed by just this experiment. Interestingly, treating yeast AAE1 with the glycosidase PNGase F that removes most N-glycans did not change the apparent AAE size in the blot. We updated the manuscript with these experiments and suggested that AAEs are likely modified by post-translational modification, but the exact modification is not clear and may be confirmed by peptide mass spectrometry (Line 250-255 of the revised manuscript).

Reviewer: I appreciate the in silico efforts and attempts to functionalize AAE1 in *E. coli*. The further experimental testing of the glycosylation hypothesis as well as the updated manuscript text on the intriguing finding of the molecular mass of AAE when expressed in yeast improves the credibility and the response is satisfactory.

Thanks very much for the reviewer's positive feedback.

- The missing VR peptides from the partial purification experiment (L235).

In both previous experiment by Geisler et al. (2016) and our experiment, the RR6 or RR6-2 protein containing precisely the "VR peptides" had no activity with vomilenine. Our best guess is

that RR6/6-2 co-purified with VR in the partial purification experiment.

Reviewer: This is plausible, and response considered sufficient.

Thanks very much for the reviewer's positive feedback.

- Intermediate stability being limiting for vomilenine biosynthesis is plausible. However, it would be of interest to the readership which intermediate would be limiting/destable. Could the authors try to truncate the pathway VOM module and perform LC-MS to verify this hypothesis (L369).

We have expressed geissoschizine synthase (GS) and sarpagan bridge enzyme (SBE) in our *de novo* yeast strain AJM7- Δ HYS, on pESC-Ura plasmid. Very low amounts of putative polyneuridine aldehyde, the SBE product, was detected only when SBE was expressed. Since we do not have a polyneuridine aldehyde standard and this MIA was known to be unstable, its detection was based on 1) comparing the MS/MS fragmentation pattern to the reference by Dang et al. (2018) Nat Chem Biol, and 2) its appearance only when SBE was expressed. The result (Supplementary Figure 16) shows that only very small amounts of polyneuridine aldehyde accumulated in yeast culture, even when encoded on high-copy plasmid. Based on this experiment and the detection of ajmaline by increasing SBE, PNAE, and VS copies, we conclude that the intermediate stability is a limiting factor for vomilenine and ajmaline production in yeast. We briefly discussed the hypothesis in Line 336-340 of the revised manuscript.

Reviewer: The AJM7- Δ HYS strain from the authors' previous study (orig. ref. 14) is indeed a relevant departure point for this study. However, there are several plausible reasons for the lack of increased vomilenine abundance beyond intermediate instability. While, it is true that Dang et al., reported low levels of polyneuridine aldehyde and 16-epivellosimine in yeast, in both Dang et al., and in this study, such low abundance could equally well be caused by shunt product formations derived from native yeast enzymes. Likewise, because genes encoding plant biosynthetic enzymes are expressed from high-copy plasmids in yeast, does not mean that these enzymes indeed will express well in yeast. Such logic should be avoided, and/or substantiated by (relative) quantification of enzyme levels (e.g. tagged-protein expression, targeted proteomics). Also, the hypothesis that "the yeast environment" is suboptimal for the stability of intermediates between geissoschizine and ajmaline is an overstatement without further analysis on this hypothesis. Further to this, this reviewer does not see that over-expression analysis of SBE, PNAE, and VS leads to increased ajmaline biosynthesis - this is only obtained by over-expression of VR-DHVR (Fig. 5B),

In conclusion, this reviewer does not find the argumentation for the low vomilenine abundance to be fully justified based on Dang et al., and by over-expression analysis of SBE, PNAE, and VS, and would like to see more experimental evidence before this hypothesis can stand alone. Alternatively, but less justified considering the target journal/audience, would be to discuss further the observed limited abundances of vomilenine by considering the alternative reasoning from this reviewer.

We thank the reviewer for thoughtful comments again. The reviewer is correct that multiple reasons could all or separately contributed to the observed marginal production of polyneuridine aldehyde and reduced pathway flux leading to vomilenine accumulation in our yeasts. In the updated results, we looked at the protein levels of SBE and PNAE by Western blot, and the

accumulation of polyneuridine aldehyde and its possible degradation products by LC-MS.

As shown in Supplementary Figure 18 below, the expression of GS, GsSBE, and PNAE (with a C-terminal cMyc tag) on multi-copy pESC vectors showed higher expression levels than GsSBE and PNAE expressed as single genomic copy in strain AJ2B. In addition, both SBE and PNAE expression levels were enhanced in the final ajmaline-producing strain AJ6 (1 copy of each *RsSBE1*, *RsSBE2*, and *GsSBE* and 3 copies of *PNAE* integrated into genome) when compared with its parental strain AJ2B (1 copy of *GsSBE* and 1 copy of *PNAE* integrated into genome). Because GS integrated into the yeast genome was not fused with a C-terminal cMyc-tag, no protein band for GS was shown on the western blot image (lanes to the right of the protein ladder).

Previously in our fermentation experiments, both AJ2B (1x GsSBE, 1x PNAE) and AJ2C (1x *RsSBE2*, 1x PNAE) accumulated comparable amounts vomilenine (Figure 4c), although AJ2C has higher expression level of SBE. These results suggested that increasing SBE expression has minimal effect on downstream vomilenine accumulation.

Supplementary Figure 18. Western blot analysis of the expression of C-terminal cMyc-tagged GS, SBE, and PNAE in different yeast strains. The details of each yeast strain can be found in Supplementary Figure 13. Both SBE and PNAE expression levels were enhanced in the final ajmaline-producing strain AJ6 (1 copy of each *RsSBE1*, *RsSBE2*, and *GsSBE* as well as 3 copies of *PNAE*) when compared with its parental strain AJ2B (1 copy of *GsSBE* and 1 copy of *PNAE*). While AJ2C had higher SBE expression level, both AJ2B and AJ2C

accumulated comparable amounts of vomilenine (Figure 4c). Expressing C-terminal cMyc-tagged GS, GsSBE, and PNAE in the parental AJM7- Δ HYS strain on multi-copy pESC vectors showed higher expression levels than GsSBE and PNAE expressed as single genomic copy in strain AJ2B. GS integrated into the yeast genome did not fuse the C-terminal cMyc-tag, thus strains AJ2B, AJ2C, and AJ6 did not show a protein band for GS. The lower panel shows a replicate SDS-PAGE gel for protein loading.

Supplementary Figure 17. LC-MS/MS analysis for fermentation sample (in YP medium) of yeast strains with ajmaline biosynthetic pathway genes integrated into the yeast genome. **(a)** MRM m/z 351>166 was used for detecting the putative polyneuridine aldehyde. **(b)** SIM m/z 365 was used for detecting the oxidative product of polyneuridine aldehyde (a hemiacetal). **(c)** SIM m/z 247 was used for detecting the degradation product of polyneuridine aldehyde (an aromatized MIA flavopepeirine). When a single copy of *GS*, *GS-GsSBE*, and *GS-GsSBE-PNAE* were integrated into the genome of AJM7- Δ HYS, respectively, LC-MS/MS profiles of these strains showed almost no difference, indicating no accumulation of putative polyneuridine aldehyde or its oxidation and degradation products.

To further address the reviewer's concern on the difference in expressing plant biosynthetic enzymes using high-copy plasmids or genome integration in yeast, we constructed yeast strains with a single copy of *GS*, *GS-GsSBE*, and *GS-GsSBE-PNAE* integrated into the genome of AJM7- Δ HYS strain. With these new yeasts, we looked at the levels of putative polyneuridine aldehyde, and its possible degradation products. Ahamada et al. (2016) pointed out the instability of polyneuridine aldehyde, and discovered that polyneuridine aldehyde spontaneously degrades to flavopepeirine (SIM *m/z* 247) and a hemiacetal (SIM *m/z* 365). Other studies on polyneuridine aldehyde structure determination relied on reducing it to more stable alcohol polyneuridine [Edwankar et al. (2008), Turpin et al. (2020)].

As shown in Supplementary Figure 17, we found no obvious difference in the LC-MS/MS profiles of these strains, indicating no accumulation of polyneuridine aldehyde (MRM *m/z* 351>166) or its oxidation product (a hemiacetal, SIM *m/z* 365) and degradation product (an aromatized MIA flavopepeirine, SIM *m/z* 247). We could not find any other alkaloids accumulating in these strains, other than tetrahydroalstonine and those unknown *m/z* 353 products detected in supplementary Figure 15 and 16.

With the updated western blot and LC-MS results, we found that both the plasmid expression and genome expression of the VOM module genes led to similar results on the low or no detection of SBE/PNAE products. However, these intermediates must have been made, since our AJ2-4 strains (*GS-SBE-PNAE-VS*) all accumulated $\sim 100 \mu\text{g L}^{-1}$ vomilenine. Marginal amounts of putative polyneuridine aldehyde were only detected when the genes were encoded on pSEC plasmids, and it was not detected when the genes were integrated in genome. Considering the no detection of these intermediates or possible shunt products from them in our yeasts, and the results that vomilenine could accumulate to appreciable amounts, it is likely that these intermediates degraded spontaneously or converted by yeast enzymes to molecules that could not be detected by LC-MS method.

As suggested by the reviewer not to make overstatement, we revised our claim as “the intermediate stability was one of major limiting factors for vomilenine production in yeast” (Line 426 of the revised manuscript). We also added “However, we could not conclude whether these intermediates degraded spontaneously or they were converted by yeast native enzymes.” at line 429-430.

As for the statement “overexpression of SBE, PNAE, and VS leads to increased ajmaline biosynthesis” or “the detection of ajmaline by increasing SBE, PNAE, and VS copies”, we came to the conclusion based on the results for AJ5H and AJ6: while AJ5H failed to synthesize ajmaline, two additional integrated copies of *SBE*, *PNAE*, and *VS* in AJ6 enabled de novo biosynthesis of ajmaline (Line 347-350 of the revised manuscript). Overall, multiple enzymatic steps are likely responsible for the low pathway flux observed. The final de novo ajmaline production was contributed by increasing gene copies of both VOM and AJM modules.

In summary, we provided the additional experiments in Supplementary Figure 17 and 18, as well as briefly discussed the corresponding results in Line 343-346 of the revised manuscript. In addition, we provided more reference reports in Line 330-336 of the revised manuscript to support the stability concerns of the pathway intermediates. In Line 422-433, we provided more discussion

on the low flux in VOM module. Please find them below. We hope the new results and discussion are helpful for readers to understand our work.

Line 330-354

“Previous studies showed that the SBE product polyneuridine aldehyde and the PNAE product 16-epivellosimine were highly unstable^{16,23,25,47–49}. Polyneuridine aldehyde in solution spontaneously oxidizes to a hemiacetal or degrades to an aromatized MIA flavopepeirine⁴⁹, and in several cases its structural determination relied on quickly reducing it to the corresponding alcohol (polyneuridine)^{16,47}. On the other hand, 16-epivellosimine spontaneously and rapidly epimerizes to its 16-epimer vellosimine, which could enter the sarpagine biosynthetic pathway^{25,48}. We carefully analyzed the fermentation samples of AJ5H and detected the accumulation of tetrahydroalstonine (m/z 353, the by-product of GS, $83 \mu\text{g L}^{-1}$) and other m/z 353 unknown by-products at high levels (Supplementary Figure 15). We suspected that the instability of SBE and PNAE products might have been responsible for decreased pathway flux and accumulation of by-products. When only GS and SBE were overexpressed in AJM7- Δ HYS yeast strain on multi-copy pESC vectors, putative polyneuridine aldehyde was detected in yeast culture, based on the reported MS/MS fragments²³. However, the amounts lagged significantly behind other by-products such as THA and other unknown MIAs (Supplementary Figure 16). When integrated a single copy of *GS*, *GS-GsSBE*, and *GS-GsSBE-PNAE* into the genome of strain AJM7- Δ HYS, LC-MS/MS profiles of these strains showed almost no difference, indicating no accumulation of polyneuridine aldehyde or the two known oxidation and degradation products (Supplementary Figure 17). Thus, we additionally integrated two copies of *SBE*, *PNAE*, and *VS* into the genome of strain AJ5H to construct the final strain AJ6 (Table 2 and Supplementary Figure 13), to strengthen the downstream pathway of 19E-geissoschizine and redirect metabolic flux from other by-products to our target product. LC-MS/MS spectrum in Figure 5a showed the production of ajmaline with a titer of $\sim 57 \text{ ng L}^{-1}$ in strain AJ6 from simple carbon sources, representing the first report on *de novo* biosynthesis of ajmaline in a heterologous host. Western blot analysis using anti-cMyc antibodies confirmed the enhanced SBE and PNAE expressions in AJ6 strain (3 copies) when compared with its parental AJ2B strain (single copy) (Supplementary Figure 18).

Line 412-461

“After increasing the gene copy numbers of *SBE*, *PNAE*, *VS*, *VH*, *VR*, and *tDHVR*, we achieved *de novo* biosynthesis of ajmaline at a titer of 57 ng L^{-1} in *S. cerevisiae* for the first time. However, the titer was far from industrial applications, indicating the requirement of further pathway optimization using protein engineering and metabolic engineering approaches. Previously in our yeast AJM7- Δ HYS background that produces strictosidine aglycone, we recorded over $200 \mu\text{g L}^{-1}$ catharanthine production after single genomic integration of 8 catharanthine biosynthetic genes¹⁴. An extra genomic copy of *GS* more than doubled the catharanthine titer to over $500 \mu\text{g L}^{-1}$, suggesting that geissoschizine production was a major limiting factor for catharanthine titer¹⁴. In this same yeast background, we could only achieve $\sim 100 \mu\text{g L}^{-1}$ vomilenine with single genomic integration of 5 genes, and additional copies of *GS*, *SBE*, *PNAE*, *VS* or *VH* did not lead to any increase of vomilenine production (Figure 4d). Western blot showed considerably higher SBE expression in AJ2C than AJ2B, however both strains accumulated comparable amounts of vomilenine (Figure 4c and Supplementary Figure 18). With the detection of only marginal amounts of polyneuridine aldehyde (Supplementary Figure 16 and 17), our results suggested that the intermediate stability was one of major limiting factors for vomilenine production in yeast. Our finding is consistent with poor polyneuridine aldehyde and 16-epivellosimine stability in literatures^{25,47,49} and the need of coupled SBE-PNAE-VS reaction to form the more stable intermediate vinorine²³. However, we could not conclude whether these intermediates degrades spontaneously or they were converted by yeast native enzymes. In our *de novo* ajmaline producing strain AJ6, the highest accumulated MIA was tetrahydroalstonine ($63 \mu\text{g L}^{-1}$) and several unknown MIAs, which showed that most of ajmaline biosynthetic intermediates did not accumulate during continuous yeast fermentation (Supplementary Figure 16 and 17). There may be auxiliary proteins or other mechanism in *R. serpentina* plant responsible for proper vomilenine accumulation. It is also possible to improve the biosynthesis by employing a scaffold-based pathway optimization strategy, which can co-localize the pathway enzymes to channel the pathway intermediates and minimize the loss of metabolic fluxes.

Downstream of vomilenine, both VR and tDHVR have moderate substrate affinity (K_M $42 \mu\text{M}$ and $32 \mu\text{M}$

respectively, Table 1), which likely also contributed to the low metabolic flux, especially when vomilenine accumulated at $\sim 100 \mu\text{g L}^{-1}$ ($0.29 \mu\text{M}$) in yeast media. Feeding yeast with 5 mg L^{-1} ($14.5 \mu\text{M}$) led to $46 \mu\text{g L}^{-1}$ ajmaline production indicated that high levels of vomilenine production was required for downstream ajmaline accumulation (AJ4B, Figure 5b). We did not observe accumulations of any intermediates downstream of vomilenine, which suggested that these intermediates may not be stable in yeast environment and/or the VR and DHVR steps were rate limiting. Adding additional copies of both VR and tDHVR led to over 2-fold increase in ajmaline production to $96\sim 128 \mu\text{g L}^{-1}$ (Figure 5b, fed with 5 mg L^{-1} vomilenine), further confirmed that VR and DHVR were the rate limiting step in ajmaline biosynthesis in yeast. Considering all the results, ajmaline biosynthetic flux was hampered by multiple enzymatic steps in yeast. The *de novo* production of ajmaline in the final strain AJ6 is contributed by the increased expression levels of multiple enzymes, especially PNAE, VR, and tDHVR. It may be beneficial to engineer VR and DHVR or mining of their homologs in other genomes with better catalytic performance in subsequent studies.”

Are there any flaws in the data analysis, interpretation and conclusions? Do these prohibit publication or require revision?

No, there are no flaws in this study.

Thanks.

Is the methodology sound? Does the work meet the expected standards in your field?

Yes. The authors use a variety of approaches to characterise the enzymes. This reviewer would be interested to know (i.e. have the authors reason the choice) why *E. coli* was used for the reductase characterization, and then yeast for the esterase (AAE1-2) characterization.

The reason for expressing AAE in yeast is the fact that these proteins are predicted to contain the secretory signal peptides. As far as we know, none of the secretory enzymes are successfully expressed in a prokaryotic system (e.g., Berberine Bridge Enzyme in benzyloquinoline alkaloid biosynthesis and tetrahydrocannabinolic acid synthase in cannabinoid biosynthesis). Previously our group discovered and characterized another putative glycoprotein O-acetylstemmadenine oxidase (ASO) in the MIA vinblastine biosynthetic pathway, which also contains secretory signal peptides. Its functional expression was only achieved in yeast and tobacco, while *E. coli* expression did not give functional proteins. We also confirmed that AAE1 expressed in *E. coli* was inactive (line 227-228) and Supplementary Figure 11.

With these prior experiences, we assumed that AAE could not be functionally expressed in *E. coli* and we opted to use yeast expression system. The rationale and discussion on the use of eukaryotic system for AAE have been included in both the results and the discussion sections (Line 176-183 and 395-404 of the revised manuscript).

Reviewer: The reviewer thanks the authors for the clarification and regards the manuscript update satisfactory. If the authors hold negative data for functional *E. coli* expression of AAEs it would be beneficial for the readers to have it inserted in the manuscript, e.g. Suppl. Fig. 11.

Thanks for the reviewer's comment. Yes, we have provided the negative data for functional expression of AAEs in *E. coli* in Supplementary Fig. 10 and Supplementary Fig. 11. While we were able to detect the expression of RsAAE1 using western blot (Supplementary Fig. 10c), the *E. coli* expressed protein was not functional in converting 17-O-acetylnorajmaline to norajmaline

(Supplementary Fig. 11, the bottom panel).

Is there enough detail provided in the methods for the work to be reproduced?

Yes.

Minors:

L287/L300: Yield should not be presented as $\mu\text{g/L}$. This is the titer.

We thank the reviewer for this point. We have changed “yield” to titer throughout the whole manuscript (such as Line 287, 307, 321, 408, and 412)

Reviewer: Good.

Thanks.

Reviewer #2 (Remarks to the Author):

The authors have sufficiently addressed all suggestions and comments. I recommend publication.

Thanks very much for the reviewer’s positive feedback.

Reviewers' Comments:

Reviewer #1:

Remarks to the Author:

The authors have satisfactorily addressed this reviewer's concern related to the pathway flux hypothesis originally proposed by the authors. The copy number analysis and further metabolite data considerations now make for a more nuanced and compelling conclusion from the pathway refactoring in yeast.

The study is now suitable for publication.